# General trends in the calnexin-dependent expression and pharmacological rescue of clinical CFTR variants

Austin Tedman[1], John A Olson III[2,3], Minsoo Kim[2,3], Catherine Foye[4], JaNise J Jackson[4], Eli F McDonald[2], Andrew G McKee[5], Karen Noguera[5], Charles P Kuntz[1], Jens Meiler[2,6], Kathryn E Oliver[4], Lars Plate[2,7], Jonathan P Schlebach[1]*

[1]The James Tarpo Jr. and Margaret Tarpo Department of Chemistry, Purdue University, West Lafayette, United States; [2]Department of Chemistry, Vanderbilt University, Nashville, United States; [3]Program in Chemical and Physical Biology, Vanderbilt University, Nashville, United States; [4]Department of Pediatrics, Emory University School of Medicine, Atlanta, United States; [5]Department of Chemistry, Indiana University, Bloomington, United States; [6]Institute for Drug Discovery, Leipzig University, Leipzig, Germany; [7]Department of Biological Sciences, Vanderbilt University, Nashville, United States

*For correspondence:
jschleba@purdue.edu

Competing interest: The authors declare that no competing interests exist.

## eLife Assessment

This **important** study advances our understanding of how cellular quality control machinery influences cystic fibrosis (CF) drug responsiveness by systematically analyzing the effects of the chaperone calnexin on more than two hundreds of CFTR (cystic fibrosis transmembrane regulator) variants. The evidence supporting the conclusions is **convincing**, with a comprehensive deep mutational scanning methodology and rigorous quantitative analysis. The findings reveal that calnexin is critical for both CFTR protein expression and corrector drug efficacy in a variant-specific manner, providing invaluable insights that could guide the development of personalized CF therapies. This work will be of significant interest to researchers in protein folding, CF drug development, and genetic disease therapeutics.

**Abstract** Cystic fibrosis (CF) is a genetic disease caused by mutations in the cystic fibrosis transmembrane conductance regulator gene (*CFTR*). Though most people with CF have one or two copies of the ΔF508 mutation, there are hundreds of other distinct CF mutations that vary in their mechanistic effects and response to therapeutics. Endogenous chaperones are known to have divergent effects on the druggability of CF variants. Nevertheless, it remains unclear how this proteostatic modulation is related to the underlying mechanistic effects of distinct classes of CF mutations. Here, we survey the effects of a previously discovered effector (calnexin, CANX) on the expression and pharmacological rescue of 232 CF variants using deep mutational scanning. We find that CANX is generally required for robust plasma membrane expression of the CFTR protein, particularly for CF variants that perturb its second nucleotide-binding domain. CANX also appears to be critical for the pharmacological rescue of CF variants with poor basal expression. Though corrector selectivity is generally dictated by the properties of mutations, we find that CANX enhances the sensitivity of CF variants within a domain-swapped region of membranes spanning domain 2 to the type III corrector VX-445. Overall, mutagenic trends suggest CANX modulates the later stages of CFTR assembly and disproportionately affects variants bearing mutations within the C-terminal domains. Interestingly,

we find that the loss of CANX results in widespread perturbations of CF variant interactomes and that the proteostatic effects of CANX are generally decoupled from changes in CFTR activity. Together, our findings reveal how the proteostasis machinery may shape the variant-specific effects of corrector molecules.

## Introduction

Cystic fibrosis (CF) is a genetic disorder affecting ~100,000 patients worldwide that is caused by mutations that impair the expression and/or function of a chloride channel known as the Cystic Fibrosis Transmembrane Conductance Regulator (CFTR). Over 1700 CF-causing loss-of-function mutations have been identified in the *CFTR* gene to date (https://www.cff.org). These mutations cause a wide variety of molecular defects ranging from the disruption of transcription and splicing of the CFTR transcript to functional defects in the channel protein (*Veit et al., 2016a*). However, most CF mutations enhance CFTR misfolding, which promotes aberrant interactions with endogenous molecular chaperones that trigger its retention and premature degradation within the endoplasmic reticulum (ER) (*Cheng et al., 1990*). The folding defects caused by certain CF variants can be suppressed by small molecule 'correctors' that bind and stabilize the CFTR protein. Two such corrector molecules (VX-661 and VX-445) are currently used in combination with an activating 'potentiator' molecule (VX-770) in the drug cocktail Trikafta, which has been approved for the treatment of numerous patient genotypes (*Middleton et al., 2019*; *Taylor-Cousar et al., 2017*). However, the underlying reasons why many clinical CF variants do not respond to these and other emerging CFTR modulators remain unknown. Efforts to develop targeted next-generation CF therapeutics will require new approaches to efficiently profile the sensitivity of clinical CF variants to emerging CFTR modulators (i.e. theratype) (*McKee et al., 2023*; *Veit et al., 2021*).

Ongoing efforts to rationalize the molecular basis for the diverse responses of clinical CF variants to various CFTR modulators have revealed that pharmacological responses coincide with changes in the CFTR interactome (*Iazzi et al., 2022*; *Louie et al., 2012*; *McDonald et al., 2022a*; *Kim et al., 2023*). Misfolded CFTR variants typically form numerous aberrant interactions with cellular quality control (QC) proteins, many of which are lost upon successful rescue with CFTR modulators (*McDonald et al., 2022a*; *Kim et al., 2023*). These perturbations of the CFTR interactome vary considerably across CF variants with distinct theratypes. Such changes in the interactome of CF variants may therefore serve as a molecular 'fingerprint' that links theratypes to differences in the underlying mechanistic effects of both the mutations and modulators. However, mass-spectrometry-based interactome measurements cannot be readily scaled to survey the entire spectrum of clinical CF variants or how they are remodeled by the growing array of clinical and emerging pre-clinical CFTR modulators. In contrast, our recent development of deep mutational scanning (DMS) approaches has enabled the characterization of hundreds of clinical CF variants in parallel (*McKee et al., 2023*). In this work, we leverage findings from recent interactome measurements in conjunction with Cas9-mediated gene editing and DMS to profile the effects of a key interactor on the expression and pharmacological response of 232 clinical CF variants.

Recent reports have identified several interactions that appear to be intimately connected to the success or failure of therapeutic intervention. While the loss of late-stage autophagocytic degradation interactions represents a common indicator of the successful correction of CFTR misfolding, there are also decisive interactions that occur within the early secretory pathway that play a more direct role in CFTR assembly and folding (*Kleizen et al., 2021*; *van der Sluijs et al., 2024*; *Luciani et al., 2010*). One such example is calnexin (CANX), which is an integral membrane lectin chaperone that recognizes misfolded proteins bearing certain glycosyl modifications (*McDonald et al., 2022a*; *Glozman et al., 2009*) CANX is a central component of the lectin chaperone cycle that recognizes improperly folded *N*-glycosylated proteins and facilitates their sorting between ER export and degradation pathways (*Kozlov and Gehring, 2020*). CANX's lectin domain recognizes mono-glucosidated *N*-glycans while the proline-rich C-terminal domain recruits various QC proteins to rectify folding defects (*Kozlov and Gehring, 2020*; *Lamriben et al., 2016*). The subsequent modification of its client proteins by glucosidases then determines which clients will be exported from the ER in order to avoid ubiquitination and degradation (*Kozlov and Gehring, 2020*; *Lamriben et al., 2016*). Recent interactome measurements revealed that the CF mutants ΔF508 and P67L enhance the propensity of

CFTR to interact with CANX (*McDonald et al., 2022a*). This aberrant interaction is partially reversed upon rescue of P67L with the corrector VX-809 (*McDonald et al., 2022a*). In contrast, ΔF508 is generally insensitive to VX-809 and treatment with this corrector does not suppress CANX interactions. Though CANX was not found to substantially alter ΔF508 expression, it is required for proper WT CFTR biosynthesis (*Farinha and Amaral, 2005*; *Okiyoneda et al., 2008*). Furthermore, CANX inhibitors disrupt the native CFTR assembly pathway (*Rosser et al., 2008*). Regardless of whether CANX interactions modulate misfolding or sorting within the early secretory pathway, this interaction may serve as a useful marker for correctability among CF variants. Nevertheless, it remains unclear whether the nature of this interaction is generalizable across the whole spectrum of clinical CF variants.

In this work, we use Cas9-mediated gene editing in conjunction with DMS to survey how CANX interactions modulate the expression and corrector response of 232 clinical CF variants. Our results reveal that knocking out CANX generally reduces the plasma membrane expression (PME) of CFTR, but disproportionately impacts the expression of variants bearing mutations within its C-terminal domains. The loss of CANX also compromises the rescue of poorly expressed CF variants and modifies the corrector selectivity of variants bearing mutations within the domain-swapped region of membrane spanning domain 2 (MSD2). Interestingly, we find drastic changes in the expression of these variants generally appear to be decoupled from changes in CFTR conductance, which suggests CANX interactions in the early secretory pathway ultimately increase the expression of CFTR but decrease its specific activity at the plasma membrane. Finally, we show that the loss of CANX interactions is accompanied by widespread changes in the CFTR interactome, including interactions associated with translational regulation. Together, our results provide new insights into how the proteostasis network modulates the variant-specific effects of correctors and vice versa.

## Results

### Expanding and benchmarking a DMS library for CFTR2 variants

To ensure comprehensive coverage of known CF variants, we first expanded our recently described DMS library by incorporating 106 additional plasmids encoding missense or small insertion/deletion variants contained within the CFTR2 database (*McKee et al., 2023*). Briefly, we used site-directed mutagenesis to introduce each mutation into a previously described DMS expression vector containing a randomized stretch of 10 bases in the plasmid backbone and a CFTR cDNA bearing three extracellular hemagglutinin (HA) epitope tags in extracellular loop four. We then used whole plasmid sequencing to confirm the sequence of one clone for each variant and determined its corresponding 10-base unique molecular identifier (UMI). We then stoichiometrically pooled the vectors encoding these variants with the 129 other clones in our recently described variant library. The final library contains identifiable UMIs that correspond to 232 CF variants. The variants contained within this expanded library and their associated DMS measurements can be found in *Supplementary files 1 and 2*.

To evaluate the fidelity of this expanded library and measure the PME of these new variants, we employed DMS to profile the surface immunostaining of these variants as previously described (*McKee et al., 2023*). Briefly, we generated a pool of stably recombined HEK293T cells in which each individual cell expresses a single CFTR2 variant (see *Methods*). We then fractionated these cells according to the relative surface immunostaining of their expressed CFTR variants, extracted the genomic DNA from each cellular fraction, and applied deep sequencing to track the relative abundance of each variant within each fraction. PME levels were then calculated from the sequencing results as previously described (*Penn et al., 2020*). PME measurements determined from this expanded library were highly correlated with our recently reported measurements for the 128 variants included in the first generation DMS library (Pearson's $R^2$ = 0.9975, *Supplementary file 2*, *Figure 1—figure supplement 1*), which confirms the integrity of the scoring is maintained in the second-generation library (*McKee et al., 2023*). The PME intensities of the new 106 variants are generally in line with our previously reported trends: a greater proportion of the mutations in membrane spanning domain 1 (MSD1) compromise CFTR expression relative to those in C-terminal domains of CFTR (*Supplementary file 2*). Overall, this expanded library offers a more comprehensive coverage of the CFTR2 database with no noticeable difference in the fidelity of DMS scoring.

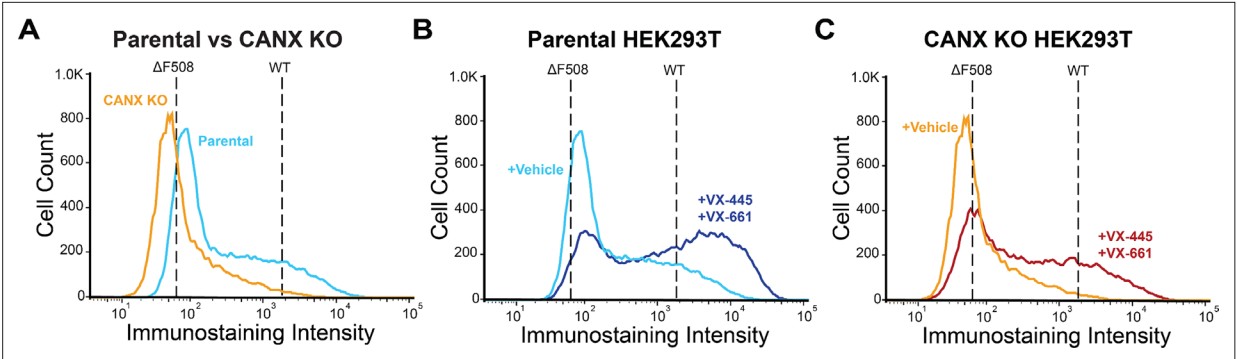

**Figure 1.** Surface immunostaining profiles of cystic fibrosis (CF) variant libraries. Flow cytometry was used to characterize the distribution of CFTR surface immunostaining intensities among recombinant CANX KO and parental HEK293T cells expressing a pool of 232 CF variants. (**A**) A histogram depicts the distribution of CFTR surface immunostaining intensities among recombinant CANX KO (orange) and HEK293T (blue) cells treated with vehicle (DMSO). The mean fluorescence intensities of parental HEK293T cells stably expressing WT or ΔF508 CFTR are shown for reference. (**B**) A histogram depicts the distribution of CFTR surface immunostaining intensities among recombinant parental HEK293T cells expressing the CF variant library treated with vehicle (light blue) or with 3 μM VX-661 + 3 μM VX-445 (dark blue). The mean fluorescence intensities of parental HEK293T cells stably expressing WT or ΔF508 CFTR are shown for reference. (**C**) A histogram depicts the distribution of CFTR surface immunostaining intensities among recombinant CANX KO cells expressing the CF variant library treated with vehicle (orange) or with 3 μM VX-661 + 3 μM VX-445 (red). The mean fluorescence intensities of CANX KO cells stably expressing WT or ΔF508 CFTR are shown for reference.

The online version of this article includes the following source data and figure supplement(s) for figure 1:

**Source data 1.** PDF containing the original western blot with band indicated for CRISPR KO validation.

**Source data 2.** Original image of CANX KO validation western blot.

**Figure supplement 1.** Comparison of deep mutational scanning measurements derived from first and second generation cystic fibrosis (CF) variant libraries.

**Figure supplement 2.** Validation of Cas9-mediated CANX knockout cells.

**Figure supplement 3.** Surface and internal expression of *N*-glycan knockouts in parental and CANX KO backgrounds.

**Figure supplement 4.** Relative surface immunostaining histograms of parental (**A**) and CANX KO (**B**) cell lines with approximate binning gates indicated as dashed lines.

## Impact of CANX on CF variant expression

We previously showed that successful correction of certain misfolded CF variants often coincides with the suppression of CANX interactions. To determine how CANX impacts this wider spectrum of rare CF variants, we first utilized CRISPR to generate a stable CANX knockout (CANX KO) cell line that can be used to identify changes in variant PME by DMS (*Methods*). Briefly, we transfected the DMS-compatible HEK293T cells with a mixture of Cas9 enzyme, a fluorescent tracrRNA, and a CANX-specific guide RNA, then used fluorescence-activated cell sorting (FACS) to isolate individual clones from the population of transfected cells. We then used genomic sequencing and western blotting to identify an edited CANX KO clone that lacks the CANX protein (*Figure 1—figure supplement 2*). Consistent with previous reports (*Glozman et al., 2009*), the loss of CANX significantly reduces the expression of both ΔF508 and WT CFTR in a manner that depends on the N-linked glycosylation of the CFTR protein (*Figure 1—figure supplement 3*). Indeed, the CFTR surface immunostaining intensity of a recombinant pool of CANX KO cells expressing the CFTR2 variant library is ~69% lower than that of the corresponding pool of parental HEK293T cells (*Figure 1A*). Moreover, CANX KO cells expressing these variants appear to exhibit less sensitivity to treatment with the combination of VX-661 and VX-445 (*Figure 1B, C*)—two correctors contained within the CF therapeutic Trikafta. Consistent with previous observations, these results suggest CANX plays a major role in CFTR maturation and its response to correctors.

To identify specific variants that are particularly sensitive to CANX, we used DMS to survey the PME of the CFTR2 library, as previously described, in the context of CANX KO cells. A linear fit of the CF variant surface immunostaining intensities in CANX KO cells against their corresponding values in parental HEK293T cells has a slope of 0.26 and all CF variants exhibit decreased PME in CANX KO cells (*Figure 2A*, *Figure 2—figure supplement 1A*). Notably, there are variants

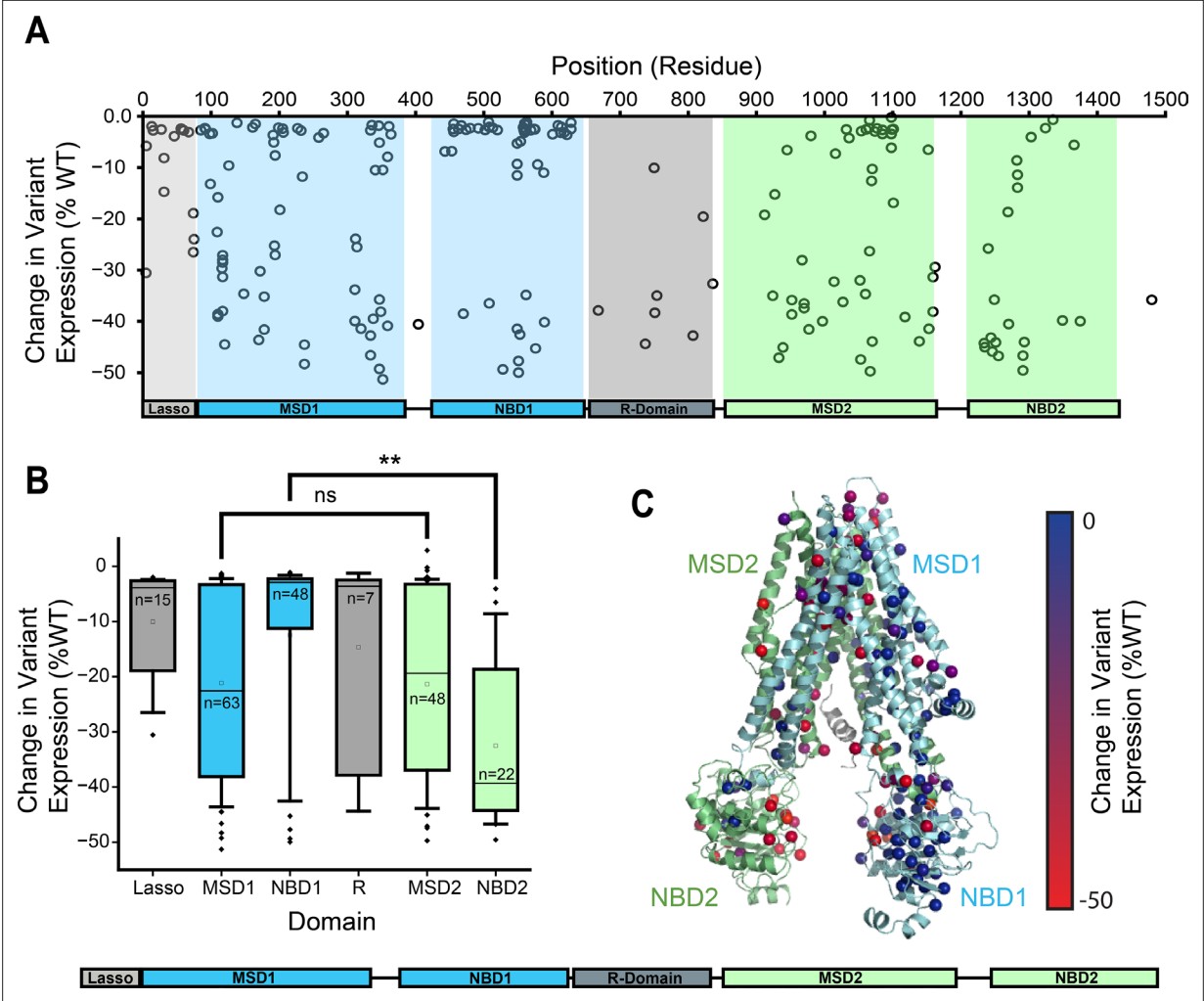

**Figure 2.** Influence of calnexin on cystic fibrosis (CF) variant plasma membrane expression (PME). The difference PME in CANX KO cells relative to the parental HEK293T cells is shown for 232 CF variants across three biological replicates. (**A**) The change in expression for each variant is plotted against its position within the CFTR sequence. The boundaries of the six CFTR domains highlighted for reference. (**B**) A box and whisker plot depicts the statistical distributions for the change in variant PME across variants within each subdomain. The upper and lower edges of the box reflect the 75th and 25th percentile values, and the upper and lower whiskers reflect the 90th and 10th percentile values, respectively. The lines within each box represent the median value and the squares within the box represent the average. ** denotes p < 0.01 for a Mann–Whitney U-test. (**C**) Values for the change in PME for each variant are projected onto their corresponding residues within a structural model of CFTR (5UAK).

The online version of this article includes the following figure supplement(s) for figure 2:

**Figure supplement 1.** Plasma membrane expression of cystic fibrosis (CF) variants in parental HEK293T cells and CANX knockout HEK293T cells.

throughout the primary structure that exhibit large decreases in PME in CANX KO cells (*Figure 2A*). However, these effects are not evenly distributed across the structural domains. While CF variants within MSDs 1 and 2 exhibit similar sensitivities to CANX overall, NBD2 variants fare consistently worse than those that fall within NBD1 (*Figure 2B*). Indeed, a projection of these variant effects onto the three-dimensional structure of CFTR suggests many of the variants that are most adversely affected are located within the C-terminal structural domains, including the domain-swapped TMDs in MSD1 that primarily form tertiary contacts within MSD2 (*Figure 2C*). Together, these trends suggest CANX is a particularly important modulator of CF variants that perturb the assembly of the C-terminal domains of CFTR. This vectorial bias potentially stems from the position of the *N*-linked glycosylation site within ECL4, which is likely to recruit CANX during the synthesis and assembly of NBD2.

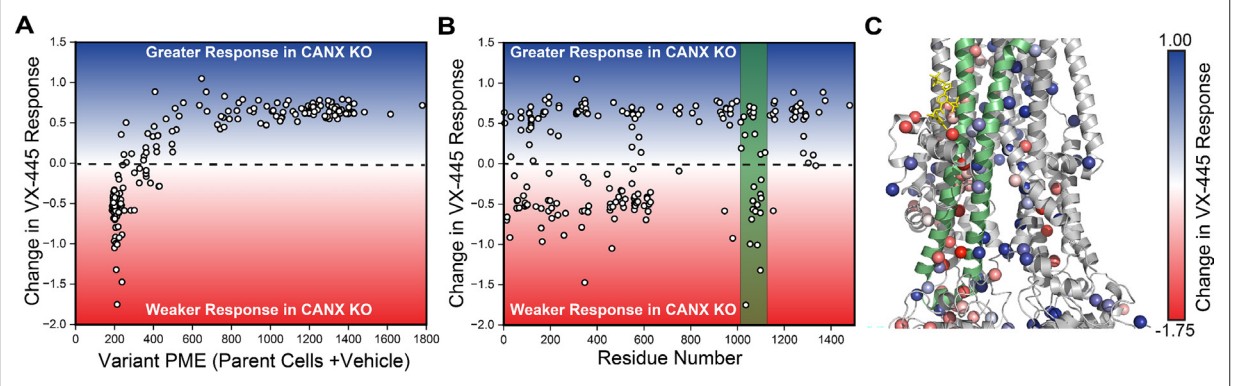

**Figure 3.** Influence of calnexin on the sensitivity of cystic fibrosis (CF) variants to VX-445. Deep mutational scanning was used to quantitatively compare the effect of 3 μM VX-445 on the plasma membrane expression (PME) of 232 CF variants across three biological replicated in CANX KO cells relative to the corresponding parental HEK293T cell line. (**A**) A scatter plot depicts the change in the VX-445 response in CANX KO cells relative to parental cells against the corresponding PME in parental HEK293T cells for each CF variant. (**B**) A scatter plot depicts the change in the VX-445 response in CANX KO cells relative to parental cells against the position of the mutated residue for each CF variant. (**C**) The change in the VX-445 response in CANX KO cells relative to parental cells for each CF mutant is projected on to the structure of VX-445-bound ΔF508 CFTR (PDB 8EIG). The structure of VX-445 is shown in yellow for reference. Negative values indicate a weaker response and positive values indicate a greater response to VX-445. Transmembrane helices 10 and 11 and ICL4 are depicted in green.

## Impact of CANX on the sensitivity of CF variants to VX-445

We previously found that the PME of many misfolded C-terminal variants could be rescued with the type III corrector VX-445 (*McKee et al., 2023*). To determine whether the enhanced CANX dependence of CF variants within the C-terminal subdomains could be reversed by VX-445, we repeated our DMS in the presence of VX-445. To quantitatively compare the response of these variants to VX-445 in each cell line, we calculated the ratio of the variant PME in the presence and absence of the drug in the context of the knockout cells and subtracted the corresponding ratio in the parental HEK293T cells (see *Methods*). A plot of the resulting change in VX-445 response for each variant against its basal expression level in the parental HEK293T cells suggests the differences in sensitivity are closely tied to basal expression (*Figure 3A*). Poorly expressed variants are generally less sensitive in the knockout lines, while those with measurable basal expression are more sensitive in the knockout line. This observation parallels findings from our previous investigation suggesting residual expression is a key factor in corrector-mediated rescue (see *Discussion*). We note that these gains in sensitivity do not reflect higher expression in the presence of the corrector but rather a larger change relative to lower basal expression in the CANX KO line (*Supplementary file 2*).

As the majority of poorly expressed CF variants reside within MSD1 and NBD1, many of the variants that exhibit diminished VX-445 sensitivity fall within the N-terminal portions of the CFTR protein (*Figure 3B*). However, there is a notable cluster of C-terminal variants that also exhibit diminished sensitivity to VX-445 (see the green stripe in *Figure 3B*). These variants fall within TMDs 10 and 11—a domain-swapped portion of MSD2 that makes structural contacts within MSD1 and forms part of the VX-445-binding site (*Figure 3C*). While many of these variants preferentially respond to VX-445 in parental cells (*McKee et al., 2023*), their rescue is compromised in the absence of CANX (*Supplementary file 2*). Computational docking studies for certain CFTR variants of interest reveal that VX-445 binding has little impact on G85E but should generally stabilize P5L, V232D, L1077P, W1098R, and N1303K CFTR to a similar degree (*Figure 4A*). While impacts on stability appear to be quite similar, modeling suggests the conformational effects of the mutations and of VX-445 binding are more nuanced. For instance, P5L, L1077P, and W1098R generate conformational defects that are allosterically propagated to NBD2 (*Figure 4B*). In each of these cases, the binding of VX-445 allosterically suppresses these defects (*Figure 4B, D*). These observations potentially suggest a means by which the loss of CANX could compromise the pharmacological rescue of variants that NBD2 misassembly—CANX and VX-445 are potentially both needed to support NBD2 assembly. However, any such perturbations of the native ensemble appear to be more subtle for other misfolded variants including G85E, V232D, and N1303K (*Figure 4A, C, E*),

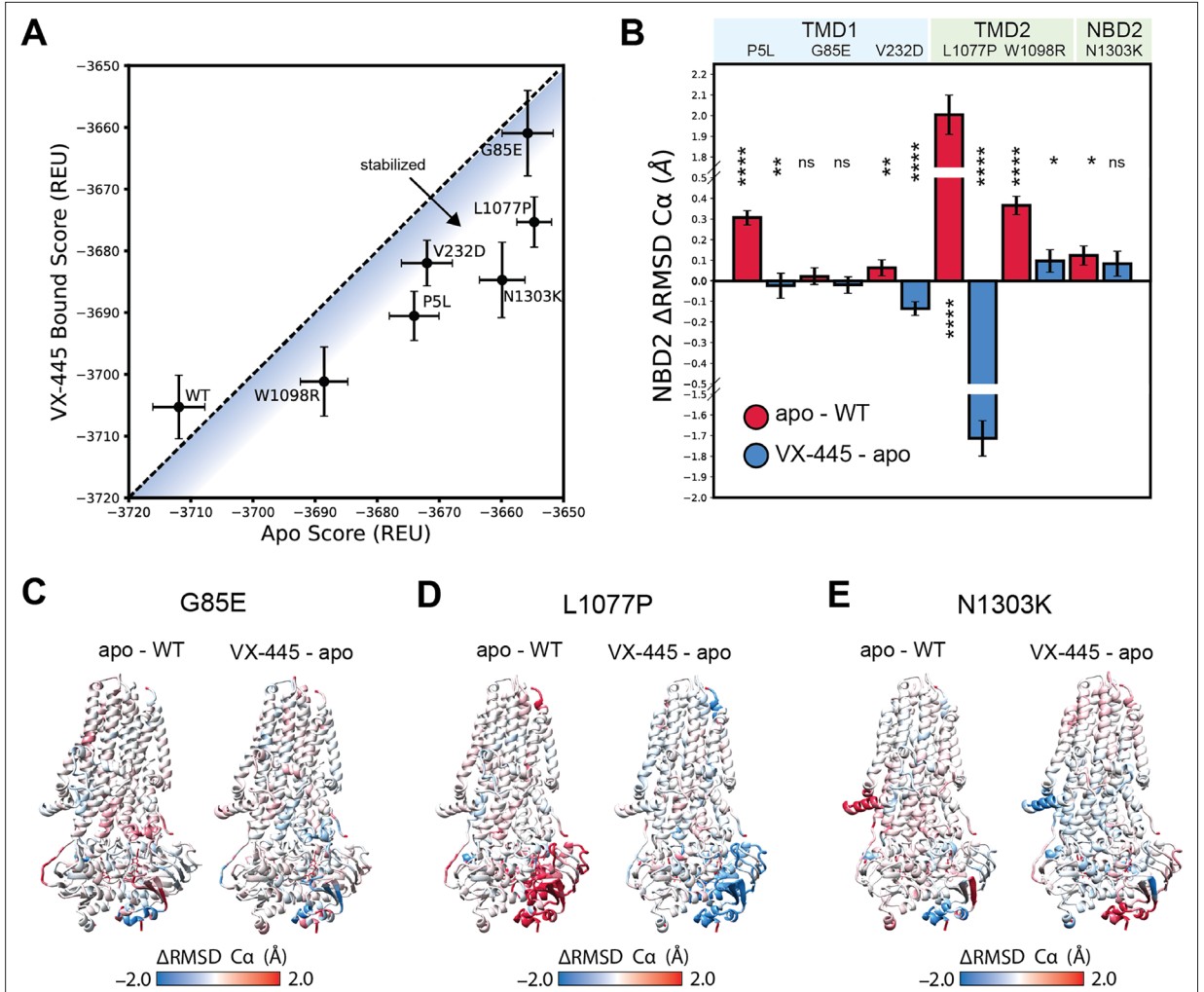

**Figure 4.** VX-445-mediated suppression of conformational defects in NBD2. Structural modeling was used to compare the conformational states of the apo and VX-445-bound active structures of WT CFTR and six rare cystic fibrosis (CF) variants (P5L, G85E, V232D, L1077P, W1098R, and N1303K). (**A**) The average Rosetta energy scores (± SEM) for the 100 lowest scoring models of the VX-445-bound state are plotted against those of the apo CFTR variant models. A reference line corresponding to no stabilization is shown for reference. All variants except for the non-responsive G85E fall below the line, which confirms VX-445 enhances stability. (**B**) The total ΔRMSD of the active conformation of NBD2 is shown for variants bound to VX-445. Red bars show increasing deviations from the native NBD2 conformation in the mutant models and blue bars how much VX-445 suppresses these conformational defects in NBD2. (**C**) Maps of the change in root mean squared deviation (RMSD) between G85E modeled with and without VX-445 show which structural regions are stabilized by VX-445. Structurally variable regions in the ensemble shown in red, while areas adopting a more ordered conformation are shown in blue. VX-445 appears to primarily suppress conformational defects within the NBD2 region of this variant. (**D**) Maps of the change in RMSD between L1077P modeled with and without VX-445 show which structural regions are stabilized by VX-445. Structurally variable regions in the ensemble are shown in red, while areas adopting a more ordered conformation are shown in blue. VX-445 appears to primarily suppress conformational defects within the NBD2 region of this variant. (**E**) Maps of the change in RMSD between N1303K modeled with and without VX-445 show that few structural regions are stabilized by VX-445 for N1303K, which responds poorly to VX-445 in vitro. Statistical significances were calculated using a non-parametric Wilcoxon signed-rank test compared to zero to determine if distributions changes were significantly different from zero, and p-values were depicted by *<0.05, **<0.01, and ****<0.0001. Error bars indicate the standard error of the mean.

which suggests this allosteric crosstalk may not be fully generalizable. Indeed, our previous findings suggest the activity of VX-445 may depend upon its interaction with nascent CFTR molecules that have yet to achieve the native structures characterized in these models (*McKee et al., 2023*). Overall, these results suggest that the loss of CANX generally hampers VX-445-mediated rescue of poorly expressed variants as well as those that compromise the assembly of the C-terminal domains.

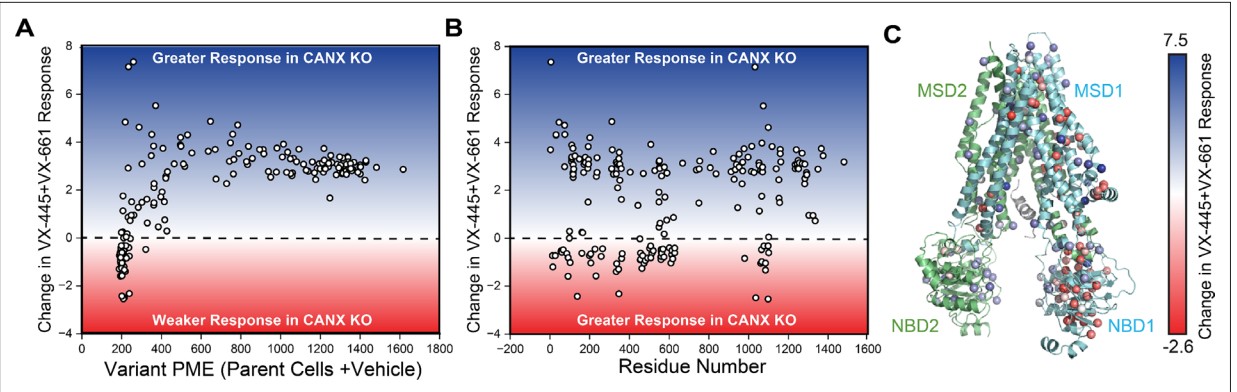

**Figure 5.** Influence of calnexin on the sensitivity of cystic fibrosis (CF) variants to VX-445 + VX-661. Deep mutational scanning was used to quantitatively compare the effect of 3 μM VX-445 + 3 μM VX-661 on the plasma membrane expression (PME) of 232 CF variants across three biological replicates in CANX KO cells relative to the corresponding parental HEK293T cell line. (**A**) A scatter plot depicts the change in the VX-445 + VX-661 response in CANX KO cells relative to parental cells against the corresponding PME in parental HEK293T cells for each CF variant. (**B**) A scatter plot depicts the change in the VX-445 + VX-661 response in CANX KO cells relative to parental cells against the position of the mutated residue for each CF variant. (**C**) The change in the VX-445 + VX-661 response in CANX KO cells relative to parental cells for each CF mutant is projected onto the WT CFTR structure (PDB 5UAK). Negative values indicate a weaker response and positive values indicate a greater response to VX-445 + VX-661.

The online version of this article includes the following figure supplement(s) for figure 5:

**Figure supplement 1.** Influence of correctors on the plasma membrane expression in parental and CANX knockout cells.

## Impact of CANX on the sensitivity of CF variants to VX-445 + VX-661

Combining VX-445 with the type I corrector VX-661 broadly enhances the rescue of misfolding CF variants (*McKee et al., 2023*; *Bihler et al., 2024*). To determine whether combining these correctors can suppress the CANX dependence of specific CF variants, we repeated our DMS in the presence of VX-445 + VX-661. Treatment with correctors generates a much smaller increase in the net PME of CANX KO cells expressing the CF variant library (*Figure 1B, C*), which clearly shows that the combination of these compounds cannot fully compensate for the loss of CANX. Similar to the variant profiles in the presence of VX-445 alone, variants with low basal PME values tend to exhibit a weaker response to this pair of correctors in CANX KO cells (*Figure 5A*). However, the gain in corrector sensitivity among variants with residual expression is greater than the degree of sensitization observed in the presence of VX-445 alone (*Figures 3A and 5A*). While this pair of correctors produces a greater overall change in expression in the CANX KO cells, the poorly expressed variants within MSD1, NBD1, and TMDs 10 and 11 still exhibit weaker response to this pair of correctors (*Figure 5B, C*). The overall similarity of the trends observed with either individual correctors or with this pair of correctors suggests any variant-specific CANX-dependent changes in variant theratype must be relatively subtle (*Figure 5—figure supplement 1*). Combining mechanistically distinct correctors does not appear to dramatically change the extent to which CF variant theratypes depend on CANX interactions.

## CANX-dependent pharmacological modulation of CF variant interactomes

Overall, our DMS measurements suggest correctors are less effective at re-balancing the cellular proteostasis of CF variants in the context of CANX KO cells. To determine how the loss of CANX modifies the effects of correctors on CF variants with divergent theratypes, we employed co-immunoprecipitation coupled to mass spectrometry to compare the interactomes of a series of CF variants that exhibit distinct dependencies on CANX (*Figure 6A*). Based on a comparison of our collective DMS measurements in the parental and knockout cells, we first compared I1366N, which exhibits an enhanced corrector response in knockout cells, to V232D, which instead has a weaker corrector response in CANX KO cells (*Supplementary file 2*). Importantly, the baseline expression of I1366N is roughly double that of V232D in parental cells and five times higher in the knockout cells (*Supplementary file 2*). Based on this consideration, it is perhaps unsurprising that V232D exhibits more pronounced increases in translation, folding, and trafficking interactions in the knockout cells (*Figure 6B*)—a larger fraction of the V232D is likely to be tied up within polysomes and/or engaged by

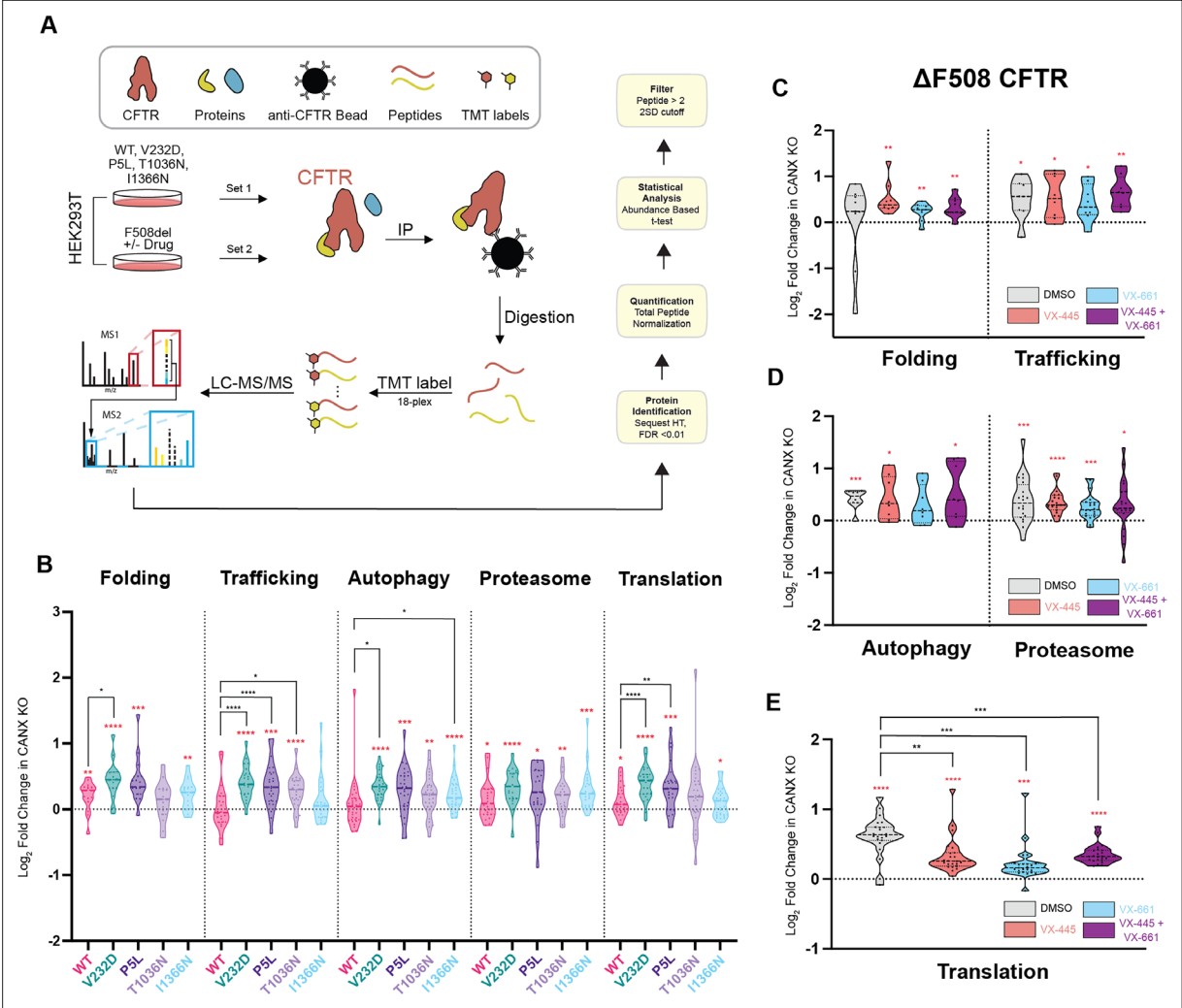

**Figure 6.** Interactome of cystic fibrosis (CF) variants in CANX KO cells. Proteomic mass spectrometry was used to compare the protein–protein interactions formed by CF variants in the context of CANX KO cells relative to the parental cell line. (**A**) A cartoon depicts the workflow for multiplexed affinity purification-mass spectrometry (AP-MS) experiments. (**B**) Violin plots depict the $\log_2$ fold change in protein abundances of various classes of interactors that associate with CF variants in CANX KO cells relative to their corresponding abundance in the parental cell line. Each data point represents individual protein interactors that are grouped according to their pathways. (**C–E**) Violin plots depict the $\log_2$ fold change in protein abundances of various classes of interactors that associate with ΔF508 in CANX KO cells treated with various correctors relative to their corresponding abundance in the parental cell line under identical treatment conditions. Each data point represents individual protein interactors that are grouped according to their pathways. Statistical significance is denoted as follows: Red asterisks indicate a significant deviation from zero, assessed using a one-sample $t$-test and Wilcoxon test against a hypothetical value of 0. Black asterisks denote significant differences from wild-type (WT) levels, determined using a repeated measures one-way ANOVA with Geisser–Greenhouse correction (*p < 0.05, **p < 0.01, ***p < 0.001, ****p <0.0001).

The online version of this article includes the following figure supplement(s) for figure 6:

**Figure supplement 1.** Comparative interactome profiling of cystic fibrosis (CF) variants in parental and CANX knockout HEK293T cells.

ER QC. Perhaps the most striking difference involves trafficking interactions. The interactions formed between I1366N and cellular trafficking proteins are quite similar in both cell lines and are indistinguishable from those of WT in the knockout cells (*Figure 6B*). These observations suggest trafficking interactions potentially play a key role in pharmacological rescue (see *Discussion*).

To expand our search for interactions that modulate theratype, we also profiled the VX-445-selective variants P5L and T1036N. Both of these variants retain minimal expression and maintain comparable corrector response profiles in each cell line, though P5L achieves higher correction overall (*Supplementary file 2*). These variants exhibit distinct perturbations in their translation and folding interactions, which suggests they are likely to misfold in distinct ways in the absence of CANX (*Figure 6B*).

Nevertheless, they each display a similar uptick in degradative interactions associated with the proteasome and autophagy pathways (*Figure 6B*), suggesting the cellular outcome of these misfolding reactions is quite similar in knockout cells. Interestingly, both variants form similar interactions with trafficking proteins, though these perturbations appear to be more similar to the severely misfolded V232D than to the I1366N variant. This may again reflect the comparable severity of the P5L, V232D, and T1036N folding defects, as these variants all fail to accumulate at the plasma membrane and/or late secretory pathway. Taken together, these collective observations suggest enhanced proteasomal interactions are associated with poor corrector response while the persistence of native trafficking interactions may facilitate greater rescue.

We previously evaluated how correctors modify the interactome of ΔF508 in HEK293T cells (*McDonald et al., 2022a*). To compare how correctors re-wire proteostatic interactions in the absence of CANX, we compared the effects of VX-661, VX-445, and VX-661 + VX-445 on the ΔF508 interactome knockout cells. We first note that ΔF508 exhibits enhanced interactions with several categories of QC proteins in the context of CANX KO cells (red stars, *Figure 6C–E*). Unlike our previous findings for ΔF508 in unmodified HEK293T cells (*McDonald et al., 2022a*), treatment with correctors does not suppress the autophagocytic and/or proteasomal interactions that mediate its degradation in the knockout cells (*Figure 6D*), which potentially explains the weaker pharmacological rescue in this cellular context (*Supplementary file 2*). However, each corrector treatment significantly suppresses the aberrant translational interactions that form in the absence of CANX (*Figure 6E*). Together, these observations suggest that the loss of CANX further compromises the cotranslational assembly of ΔF508 in a manner that cannot be fully suppressed by corrector molecules. Indeed, this observation is consistent with our recent identification of CANX as a chaperone involved in the cotranslational QC of this variant (*Carmody et al., 2024*). Nevertheless, the divergent modifications to the interactomes of the other variants in the knockout cells suggest the effects of CANX on variant theratype is likely to vary widely (*Figure 6—figure supplement 1*).

## Impact of CANX on the activity and functional rescue of CF variants

Interactome measurements suggest the loss of CANX significantly alters the manner in which the CFTR protein is assembled within the early secretory pathway. To evaluate functional consequences of this change in variant assembly and correction, we compared the activity and pharmacological response of the five divergent CF variants described above in the context of the CANX KO and the parental HEK293T cells. Briefly, we generated a series of recombinant stable CANX KO and HEK293T cell lines that inducibly express each CF variant in conjunction with a ratiometric halide-sensitive YFP (hYFP) sensor protein (*Galietta et al., 2001*; *Vijftigschild et al., 2013*) (see *Methods*). We then tracked the ratiometric quenching of hYFP fluorescence over time following CFTR activation at the single-cell level using flow cytometry, as previously described (*Carmody et al., 2024*). The observed rate of hYFP quenching is slower in the presence of the CFTR-specific inhibitor CFTR(inh)-172 (*Figure 7—figure supplement 1*), which confirms that quenching is rate-limited by CFTR conductance. As expected, quenching is faster in cells expressing WT CFTR relative to cells expressing CF variants (*Figure 7A, B*), though there are prominent differences in the activity of these variants (two-way ANOVA, p = 3.8 × 10$^{-17}$). However, the trends in quenching half-lives appear unchanged in CANX KO cells (*Figure 7B*, two-way ANOVA, p = 0.06). Thus, while DMS measurements suggest these mutations reduce the CFTR PME by an average of ~69% (*Figure 1A*), any resulting changes in functional activity appear to be quite modest. Furthermore, the CFTR modulators found in Trikafta generate a similar degree of functional rescue for each of these variants in the context of the CANX KO cells (*Figure 7C*). Overall, these findings reveal the proteostatic effects of CANX on CFTR assembly are decoupled from differences in CF variant activity, which suggests the CFTR molecules that are assembled in the absence of CANX are more active than those that undergo CANX-mediated assembly.

To determine whether CANX modulates CFTR activity in a more relevant cellular context, we measured short-circuit currents across Fischer rat thyroid (FRT) cell monolayers transiently expressing WT or ΔF508 CFTR. As expected, monolayers expressing WT CFTR exhibit robust forskolin-stimulated currents that are suppressed by CFTR(inh)-172 (*Figure 7D*). Moreover, the corresponding currents across monolayers expressing ΔF508 only reach 2.4% of the current achieved by cells expressing WT CFTR (*Figure 7E, F*). Knocking down CANX expression reduces the current in monolayers expressing WT CFTR by 27 ± 12% (*Figure 7D*, p = 0.003). Notably, this decreased CFTR activity cannot be reversed

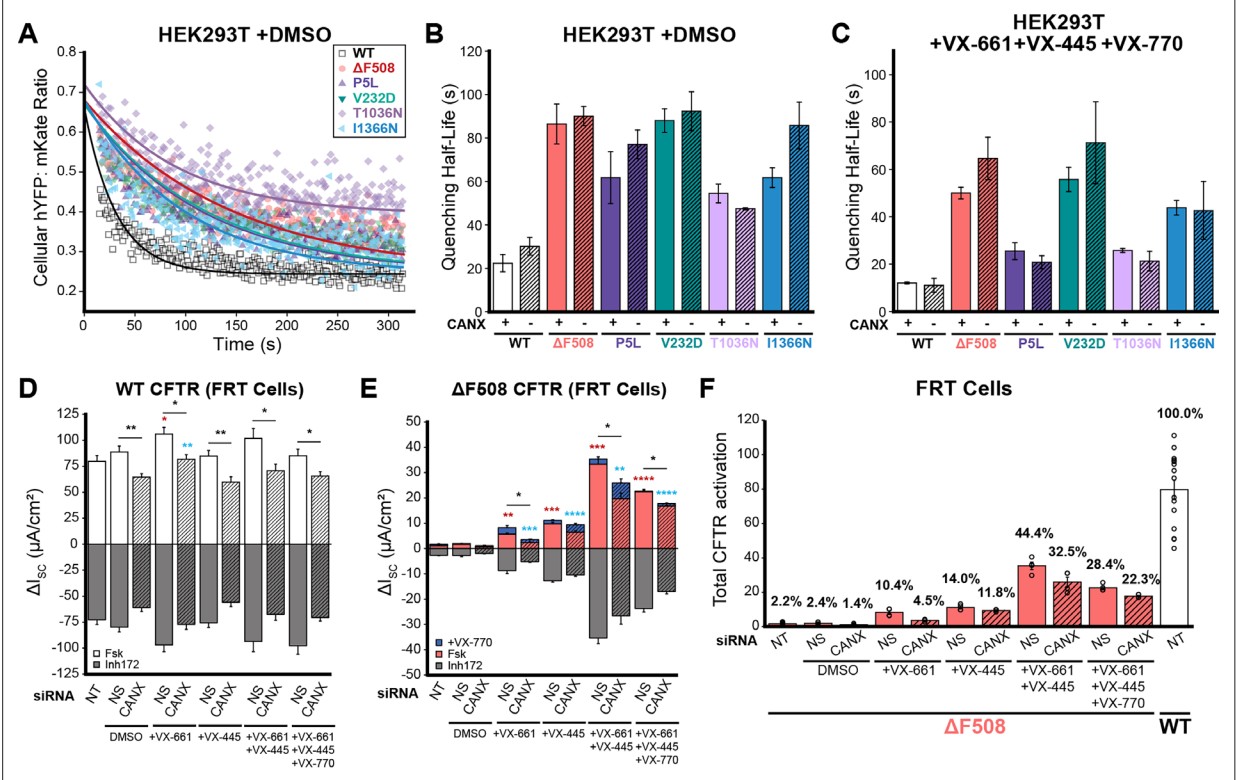

**Figure 7.** Influence of calnexin on the functional rescue of cystic fibrosis (CF) variants. The functional properties of CF variants are compared in various cells that feature endogenous CANX expression or deficient CANX expression under various experimental conditions. (**A**) Flow cytometry was used to monitor CFTR-mediated quenching of a halide-sensitive YFP (hYFP) relative to an mKate expression marker upon activation of stably expressed CFTR variants with 25 μM forskolin (Fsk) and 50 μM genistein in parental HEK293T and CANX KO HEK293T cells. Representative flow cytometry measurements of single-cell hYFP: mKate ratios are plotted over time following CFTR activation for recombinant cells stably expressing a series of CF variants. Fitted curves for single exponential fit decay are shown for reference. Bar graphs depict the fitted half-lives of the hYFP quenching reactions among cells expressing each indicated CF variant in parental HEK293T cells (CANX+, open bars) and CANX KO cells (CANX−, dashed bars) treated with (**B**) DMSO or with (**C**) 3 μM VX-661, 3 μM VX-445, and 10 μM VX-770. Values represent the mean ± SEM (*n* = 3). Fischer rat thyroid (FRT) monolayers transiently expressing (**D**) ΔF508 or (**E**) WT CFTR were cultured 5 days on permeable transwells and chronically treated with CFTR modulators (72 hr, 5 μM), vehicle control (DMSO), and/or various siRNA (72 hr, 100 nM). Bar plots depict average short-circuit currents (ΔISC) following acute application of 10 μM Fsk, 5 μM VX-770, and/or 10 μM of the CFTR-specific inhibitor-172 (Inh172). (**F**) A bar graph depicts the total functional activation of ΔF508 CFTR that is achieved under various conditions upon stimulation with Fsk and VX-770 relative to WT stimulated with Fsk. Values represent mean ± SEM (*n* = 4). Asterisks annotate statistical evaluation of Fsk ± VX-770 values compared to non-specific (NS, red) or CANX-specific (blue) siRNA. *p < 0.05; **p < 0.01; ***p < 0.001; ****p < 0.0001 (two-way ANOVA). Currents from non-treated cells (NT) are shown for reference.

The online version of this article includes the following figure supplement(s) for figure 7:

**Figure supplement 1.** Impact of a CFTR-specific inhibitor on observed hYFP quenching kinetics in recombinant cells expressing cystic fibrosis (CF) variants.

**Figure supplement 2.** Validation of siRNA-mediated CANX knockdowns in Fischer rat thyroid (FRT) cells.

by CFTR modulators (*Figure 7D*). Due to its poor basal activity, knocking down CANX appears to have little effect on ΔF508 CFTR (*Figure 7E, F*). However, knocking down CANX significantly reduces the functional rescue of ΔF508 CFTR by CFTR modulators (*Figure 7E, F*). Together, these data demonstrate that CANX does support the functional maturation and pharmacological properties of CFTR in the context of cellular monolayers. While our observations potentially suggest expression and function are more tightly coupled in FRT monolayers, we note that the average quenching half-lives in HEK293T also suggest WT CFTR is less active. The pharmacological rescue of ΔF508 CFTR is also less pronounced in the CANX KO HE293T cells, though any differences appear relatively small and are not statistically significant (*Figure 7B, C*). Thus, functional measurements appear to be qualitatively consistent across the two cell lines. We suspect the statistical differences are likely rooted in differences in methodological precision rather than meaningful functional differences. Overall, our findings

suggest that the impact of CANX on CF variant function appears modest in relation to its effect on the PME of CF variants.

## Discussion

CFTR misfolding results in the perturbation of hundreds of protein–protein interactions (*McDonald et al., 2022a*; *Kim et al., 2023*; *Pankow et al., 2015*), and the pharmacological suppression of aberrant interactions appears to be a key determinant of the sensitivity of CF variants (*McDonald et al., 2023*). Importantly, knocking down certain interactors can also suppress misfolding and enhance the pharmacological response (*McDonald et al., 2022a*; *Kim et al., 2023*; *Hutt et al., 2018*; *Coppinger et al., 2012*; *Veit et al., 2016b*; *Oliver et al., 2019*), which suggests many of these interactions limit therapeutic rescue. Recent interactome measurements have revealed that several misfolded CF variants exhibit enhanced interactions with the lectin chaperone CANX (*McDonald et al., 2022a*; *McDonald et al., 2025*). Treatment with correctors can suppress this interaction for some, but not all CF variants (*McDonald et al., 2022a*; *McDonald et al., 2025*). Here, we employ DMS to survey how CANX modulates the expression and theratype of 232 rare CF variants. Our results reveal that knocking out CANX reduces the PME of all CFTR variants (*Figure 1*, *Supplementary file 2*), though some are more affected than others (*Figure 2*). Many of the CF variants that appear to be most dependent on CANX feature mutations within NBD2 (*Figure 2B*)—the subdomain directly C-terminal of the glycosylation site recognized by CANX. Knocking out CANX also reduces the magnitude of the rescue that is generated by VX-661 and VX-445 (*Figure 1*). Consistent with our findings in parental HEK293T cells (*McKee et al., 2023*), the fold change in CF variant PME follows a familiar pattern: poorly expressed variants exhibit smaller increases in expression, while those with residual expression appear more sensitive (*Figures 3A and 5A*). However, the enhanced sensitivity of variants with residual expression in CANX KO cells generally reflects the lower basal expression of these variants rather than a higher overall expression in the presence of correctors (*Supplementary file 2*). Thus, even in the context of this altered proteostatic environment, residual expression levels appear to be a dominant factor in CF variant theratype (*McKee et al., 2023*; *Sosnay et al., 2013*). The loss of CANX appears to generally decrease basal expression levels across the spectrum of variants, and the variants that lose their remaining residual expression in knockout cells become desensitized to correctors. Nevertheless, the change in theratype is not fully uniform, as we identify a cluster of variants bearing mutations within TMDs 10 and 11 that exhibit reduced rescue by VX-445 in CANX cells (*Figure 3B, C*). These two domain-swapped helices, which contain part of the VX-445-binding site (*Fiedorczuk and Chen, 2022*), must form their native contacts late in synthesis by virtue of their position within the primary structure. This observation is consistent with recent indications that both CANX and VX-445 modulate the late stages of CFTR assembly (*McKee et al., 2023*; *Carmody et al., 2024*). Together, these shed light on the nuanced ways that changes in the proteostasis environment can modulate CF variant theratype.

In an effort to determine how the proteostasis network adapts to the loss of CANX, a central component of the ER lectin chaperone cycle, we characterized the interactome of several variants that exhibit changes in theratype within knockout cells. Across five CF variants, we observed consistent upticks in a range of interactors that are involved in protein translation, folding, trafficking, and degradation (*Figure 6*). Though variations in the interactome are slightly different among these variants (*Figure 6—figure supplement 1*), our findings generally suggest that the loss of CANX has broad effects that ripple across the proteostasis network. Thus, it does not appear that one discrete chaperone system can fully compensate for the disruption of the lectin chaperone cycle. The impaired PME of these variants, along with the uptick in their translational, folding, and degradative interactions, implies that, without CANX, they generally fail to escape the ER and are mostly degraded prior to export. While this difference appears to be more or less uniform among the variants we characterized in CANX KO cells, there are subtle differences that appear to shape theratype. For instance, of the five CF variants we characterized, only one variant (I1366N) retains native-like trafficking interactions and residual expression (*Figure 6B*). Its avoidance of aberrant trafficking interactions perhaps stems from the fact that, unlike the other four characterized variants, measurable quantities of I1366N must persist within the late secretory pathway in route to the plasma membrane (*Supplementary file 2*). The heightened corrector response of this variant could potentially arise from differential effects of correctors within the early versus late compartments of the secretory pathway. Variants with residual

expression could generally be most sensitive to correctors if, for instance, corrector-mediated stabilization of the mature channel protein has a greater influence on proteostasis than the effects of these molecules on cotranslational assembly. Nevertheless, this interpretation remains speculative, and additional investigations are needed to assess the organelle-specific effects of correctors.

Though trends among variant interactomes and expression levels suggest CANX is critical for CFTR assembly, it is unclear how such changes in biosynthesis may impact channel function. We therefore employed both a halide-sensitive YFP quenching assay and Using chamber measurements to determine how a loss of CANX impacts channel activity in HEK293T cells and FRT monolayers, respectively. While we observe considerable differences in the function of CF variants (*Figure 7B*), the functional output of these variants is surprisingly similar in CANX KO HEK293T cells (*Figure 7C*). While decreases in WT and ΔF508 CFTR activity are slightly more pronounced in the context of FRT cell monolayers (*Figure 7D–F*), the attenuation of CFTR function still appears to be smaller than the corresponding reduction in PME of these variants. From an enzymology perspective, this observation suggests that the loss of CANX must *increase* the specific activity of the functionally mature CFTR protein. That is to say that the mature CFTR channels at the plasma membrane have a higher functional output, on average, in cells lacking CANX. One potential molecular interpretation would be that more inactive CFTR channel proteins manage to escape the ER and reach the plasma membrane with the help of CANX. Nevertheless, we note that this apparent disconnect between export and function is more pronounced in the context of HEK293T cells than in FRT cells. While these discrepancies could stem from differences in the dynamic range of the functional assays, they may also suggest the stringency of QC is more finely tuned to ion channel biosynthesis in epithelial monolayers. Nevertheless, the fact that early biosynthetic interactions can have a lasting effect on the functional behavior of mature channels hours after their assembly is quite remarkable. We note that these observations are consistent with our recent observations involving regulatory mRNA structures that alter the assembly dynamics and function of ΔF508 CFTR (*Carmody et al., 2024*). Additional investigations are needed to gain insights into the molecular basis for the distinct functional properties of CFTR channels with divergent biosynthetic reaction coordinates.

# Materials and methods

## Key resources table

| Reagent type (species) or resource | Designation | Source or reference | Identifiers | Additional information |
|---|---|---|---|---|
| Antibody | Anti-Hemagglutinin (Mouse Monoclonal) | Invitrogen | Cat# 26183-D550; RRID:AB_2533052 | 2–2.2.14, DyLight 550 conjugate, mouse monoclonal, 1:100 |
| Antibody | Anti-Hemagglutinin (Mouse Monoclonal) | Invitrogen | Cat# 740005MP647; RRID:AB_3666184 | 2–2.2.14, Alexa Fluor 647 conjugate, mouse monoclonal, 1:100 |
| Antibody | Human CFTR C-Terminus Antibody | Bio-Techne | Cat# MAB25031; RRID:AB_2260673 | Mouse Monoclonal, 6 mg antibody/ml of beads |
| Strain, strain background (*Escherichia coli*) | NEB10β | New England Biolabs | Cat# C3020K | Electrocompetent |
| Strain, strain background (*Escherichia coli*) | NEBDH5α | New England Biolabs | Cat# C2987H | |
| Chemical compound, drug | iodoacetamide | Sigma-Aldrich | Cat# I6125 | |
| Commercial assay, kit | Fugene 6 | Promega | Cat# E2691 | |
| Peptide, recombinant protein | 2X Kappa HiFi HotStart Ready Mix | Roche Diagnostics | Cat# KK2602 | |
| Chemical compound, drug | VX-661 | APExBIO | Cat# A2664 | |
| Chemical compound, drug | VX-445 | Selleckchem | Cat# S8851 | |

*Continued on next page*

*Continued*

| Reagent type (species) or resource | Designation | Source or reference | Identifiers | Additional information |
|---|---|---|---|---|
| Chemical compound, drug | VX-770 | Selleckchem | Cat# S1144 | |
| Chemical compound, drug | Forskolin | Selleckchem | Cat# S2449 | |
| Chemical compound, drug | Genistein | Selleckchem | Cat# S1342 | |
| Chemical compound, drug | Amiloride | Sigma-Aldrich | Cat# A7410 | |
| Chemical compound, drug | CFTRinh-172 | Selleckchem | Cat# S7139 | |
| Chemical compound, drug | Roche cOmplete Protease Inhibitor Cocktail | Roche Diagnostics | Cat# 11697498001 | |
| Commercial assay or kit | Lipofectamine CRISPRMAX Cas9 Transfection Reagent | Invitrogen | Cat# CMAX00008 | |
| Commercial assay or kit | ZymoPURE Plasmid Miniprep Kit | Zymo Research | Cat# D4211 | |
| Commercial assay or kit | Zymopure Endotoxin-Free Midiprep Kit | Zymo Research | Cat# D4200 | |
| Commercial assay or kit | ZR-96 DNA Clean-Up Kit (Shallow Well) | Zymo Research | Cat# D4018 | |
| Commercial assay or kit | Select-a-Size DNA Clean & Concentrator | Zymo Research | Cat# D4080 | |
| Commercial assay or kit | DNeasy Blood and Tissue Kits for DNA Isolation | QIAGEN | Cat# 69506 | |
| Commercial assay or kit | Detergent-Compatible Bradford Assay | Pierce Biotechnology | Cat# 23246 | |
| Transfected construct (*H. sapiens*) | Tet-Bxb1-BFP HEK293T | Laboratory of Doug Fowler | | Clone 37 described in *Matreyek et al., 2017* |
| Cell line (*H. sapiens*) | HEK293T for DMS CANX KO | This paper | | Clone 37 described in *Matreyek et al., 2017* |
| Cell line (*H. sapiens*) | HEK293T | ATCC | Cat# CRL-11268; RRID:CVCL_0063 | |
| Cell line (*R. norvegicus*) | Fischer rat thyroid (FRT) cells | Michael Welsh | | |
| Sequence-based reagent | Custom CFTR primer pool | Agilent Technologies Inc | | See *Supplementary file 3*. Supplemental Primer List Related to Variant and Deep Sequencing Library Production |
| Sequence-based reagent | CANX siRNA | QIAGEN | #SI02666335 | |
| Sequence-based reagent | NS siRNA | QIAGEN | #1027310 | |
| Software, algorithm | Origin 2019 | OriginLab | RRID:SCR_014212 | |
| Software, algorithm | Image Studio V. 5.2 | LI-COR Biosciences | RRID:SCR_015795 | |
| Software, algorithm | Structure Refinement from Cryo-EM maps | Wang et al. | | |
| Software, algorithm | RosettaLigand | Lemmon & Meiler | | |
| Software, algorithm | BCL ConformerGenerator | Mendenhall et al. | | |
| Software, algorithm | RosettaCM | Song et al. | | |

*Continued on next page*

*Continued*

| Reagent type (species) or resource | Designation | Source or reference | Identifiers | Additional information |
|---|---|---|---|---|
| Software, algorithm | RosettaMembrane | Barth et al. | | |
| Software, algorithm | OCTOPUS | Viklund & Elofsson | | |
| Software, algorithm | Proteome Discoverer 2.4 | Thermo Fisher | RRID:SCR_014477 | |

## Reagents

CFTR agonists and antagonists were purchased from commercial providers. Compounds included tezacaftor (VX-661; MedChemExpress, #HY-15448), elexacaftor (VX-445; Selleck Chemicals, #S885), ivacaftor (VX-770; MedChemExpress, #HY-13017), amiloride (Sigma-Aldrich, #A7410), forskolin (Selleck Chemicals, #S2449), and CFTR Inhibitor172 (Inh172; Selleck Chemicals, #S7139). For gene-specific knockdown, siRNAs were used to target calnexin (Canx; QIAGEN, #SI02666335) or a non-specific (NS) negative control (QIAGEN, #1027310). The following target sequences were employed (5′ → 3′): Canx, ACCAGGGAATCTGGAAACCAA; and NS, AATTCTCCGAACGTGTCACGT.

## Molecular biology

A pcDNA5 vector, in which we inserted CFTR, a triple HA tag in the fourth extracellular loop, an internal ribosome entry site-eGFP cassette, and a Bxb1 recombination site was used to generate a molecular library of barcoded CF variants. We first installed a randomized 10-nucleotide UMI upstream of the Bxb1 recombination site using nicking mutagenesis in place of the CMV promoter. A plasmid preparation containing WT plasmids with scrambled UMIs was used as a template for 106 individual site-directed mutagenesis reactions to generate a library of pathogenic CF variants that were found in the CFTR2 Database (https://www.cftr2.org/). Individual clones from each reaction were generated using the ZymoPURE Plasmid Miniprep Kit (Zymo Research, Irvine, CA) and deep sequenced by shotgun Illumina sequencing to confirm the sequence of each mutated open reading frame and to determine its corresponding 10 base UMI. Plasmids encoding individual variants were pooled with 129 CF variant-encoding plasmids previously described by *McKee et al., 2023* The plasmid pool was then electroporated into electro-competent NEB10b cells (New England Biolabs, Ipswitch, MA), which were then grown in liquid culture overnight and purified using the ZymoPure endotoxin-free midiprep kit (Zymo Research, Irvine, CA). Based on subsequent deep sequencing, the resulting plasmid pool contains 232 variants. The barcode sequence associated with L1065P could not be detected in the final pool. The Bxb1 recombinase expression vector (pCAG-NLS-HA Bxb1) was kindly provided by Douglas Fowler.

## Cell culture

HEK293T cells were grown in Dulbecco's modified Eagle's medium (Gibco, Grand Island, NY) containing 10% fetal bovine serum (Corning, Corning, NY) and a penicillin (100 U/ml)/streptomycin (100 μg/ml) antibiotic supplement (Gibco, Grand Island, NY) in a humidified incubator containing 5% $CO_2$ at 37°C. HEK293T cells used for interactome measurements were grown under similar conditions but with 1% L-glutamine (200 mM) supplementation. CFTR variant DNA constructs were stably recombined in modified HEK293T cells containing a genomic Tet-Bxb1-BFP 'landing pad' (*Matreyek et al., 2017*) by co-transfecting with Fugene 6 (Promega, Madison, WI) and Bxb1 recombinase expression vectors. The cells were then incubated at 33°C for 3 days and induced with doxycycline (2 μg/ml) 1 day after transfection. The cells were then incubated at 37°C for 24 hr prior to the isolation of GFP-positive/BFP-negative cells that had undergone recombination using a BD FACS Aria II (BD Biosciences, Franklin Lakes, NJ). These cells were grown in 10 cm dishes with complete media supplemented with doxycycline (2 μg/ml) until confluency when they were divided into 15 cm dishes for the three-drug trial replicates. Cells were dosed with 3 μM VX-661 and/or VX-445 16 hr prior to harvesting when applicable. Seven to twelve days post-transfection, cells were washed with 1X phosphate buffered saline (AthenaES, Baltimore, MD) and harvested with 1X0.25% Trypsin-EDTA (Gibco, Grand Island, NY). CFTR expressed at the plasma membrane of recombinant cells was labeled with a DyLight 550-conjugated anti-HA antibody (Thermo Fisher, Waltham, MA). Labeled cells were then fractionated into quartiles according to surface immunostaining intensity (*Figure 1—figure supplement 4*) using an FACS Aria Ilu fluorescence-activated cell sorter (BD Biosciences, Franklin Lakes, NJ).

At least 1 million cells from each fraction were isolated to ensure exhaustive sampling. Sorting gates for surface immunostaining were independently set for each biological replicate and in each condition to ensure that the population was evenly divided into four equal subpopulations. Fractionated subpopulations were expanded in 10 cm culture dishes prior to harvesting and freezing 15–25 million cells per quartile fraction for the downstream genetic analysis.

Parental FRT cells were a gift of Michael Welsh (University of Iowa, USA). As detailed previously (*Oliver et al., 2019*; *Sabusap et al., 2016*; *Rauscher et al., 2021*), cells were cultured in F12 Ham Coon's modified nutrient mixture (Sigma-Aldrich, St. Louis, MO) supplemented with 2.68 g sodium bicarbonate, 850 µl 2 N hydrochloric acid, and 5% fetal bovine serum (pH 7.3). Incubator conditions were set to 5% $CO_2$, 95% $O_2$, and 37°C. Parental FRT were co-transfected with plasmid DNA and siRNA based on established protocols (*Oliver et al., 2019*; *Rauscher et al., 2021*; *Ramachandran et al., 2013*; *Sabusap et al., 2021*) In short, cells were seeded on 0.33 $cm^2$ permeable transwells at $1.5 \times 10^5$ cells/well and allowed to adhere overnight. The following day, human F508del- or WT-CFTR cDNA cloned into the pcDNA5 vector (Invitrogen, Waltham, MA) was transfected (0.5 µg) onto the apical surface of FRT monolayers with 0.5 µl Lipofectamine 3000 and 0.75 µl P3000 (Invitrogen, Waltham, MA) suspended in 20 µl 1X Opti-MEM Reduced Serum (Gibco, Grand Island, NY). Plasmids incubated 24 hr, then were aspirated to facilitate apical transfection of siRNA duplexes. CANX or NS siRNA (100 nM) were delivered to polarized FRT cells by forward transfection using 0.5 µl Lipofectamine RNAiMAX (Invitrogen, Waltham, MA). Duplexes of siRNAs were incubated 24 hr, then aspirated to induce cell culture at air–liquid interface. Transfections with siRNA generated a ~50% knockdown efficiency (*Figure 7—figure supplement 2*).

## Generation of CRISPR calnexin knockout cells

A modified HEK 293T cell line containing a bxb1 genomic landing pad (*Matreyek et al., 2017*) was transfected with CANX using CRISPRMAX (Thermo Fisher, Pittsburgh, PA) and verified via western blot. Alt-R CRISPR–Cas9 crRNA complement to exon 8 was identified and ordered via IDT Genome editing. 1 µM of this crRNA was incubated with 1 µM Alt-R CRISPR–Cas9 tracrRNA at 95°C for 5 min. 6 µl of this mixture was incubated with 6 µl 1 uM Alt-R S.p. Cas9 Nuclease V3, 2.4 µl Cas9 Plus, and 85.6 µl OPTI-MEM at room temperature for 5 min. 100 µl of this RNA complex was incubated with 4.8 µl CRISPRMAX and 85.2 µl OPTI-MEM at room temperature for 20 min. While incubating, the cells were washed and resuspended using 1 × 0.25% Trypsin-EDTA, counted, and diluted to 400,000 cells/ml in DMEM complete media. 50 µl of the RNA complex and 100 µl of cell were added to each of 48 wells, incubated for 48 hr at 37°C. The cells were then resuspended and sorted for single cells on ATTO550 using an FACS Aria IIu fluorescence-activated cell sorter (BD Biosciences, Franklin Lakes, NJ) into a single well of a 96-well plate. The cells were grown up, harvested, and lysed to be analyzed by western blot and sanger sequencing.

### Quantification of variant surface immunostaining

Relative surface immunostaining calculations were estimated from sequencing data using a computational approach described previously (*McKee et al., 2023*). Briefly, low-quality reads that likely contain more than one error were first removed from the demultiplexed sequencing data. UMI sequences within the remaining reads were then counted within each sample to track the relative abundance of each variant. To compare read counts across fractions, the collection of reads within each population was then randomly down-sampled to ensure a consistent total read count across each subpopulation. The surface immunostaining of each variant was then estimated by calculating the weighted-average immunostaining intensity for each variant using the following equation:

$$< \mathrm{I} >_{Variant} = \frac{\sum_{k=0}^{4} < F >_k N_k}{\sum_{k=0}^{4} N_k} ,$$

where $\langle I \rangle_{Variant}$ is the weighted-average fluorescence intensity of a given variant, $\langle F \rangle_i$ is the mean fluorescence intensity associated with cells from the $i$th FACS quartile, and $N_i$ is the number of variant reads in the $i$th FACS quartile. Variant intensities from each replicate were normalized relative to one another using the mean surface immunostaining intensity of the entire recombinant cell population for each experiment to account for small variations in laser power and/or detector voltage. Finally,

to filter out any noisy scores arising from insufficient sampling, we repeated the down-sampling and scoring process, then rejected any variant measurements that exhibit more than 40% variation in their intensity scores across the two replicate analyses. The reported intensity values represent the average normalized intensity values from two independent down-sampling iterations across three biological replicates.

## Interactome profiling

MS sample preparation of co-IP samples was performed as described previously (*Kim et al., 2023*). Cell lysates were precleared with 4B Sepharose beads (Sigma-Aldrich, St. Louis, MO) at 4°C for 1 hr while rocking. Precleared lysates were then immunoprecipitated with Protein G beads conjugated to 24-1 antibody (6 mg antibody/ml of beads) overnight at 4°C while rocking. Resin was washed three times with a TNI buffer containing 50 mM Tris (pH 7.5), 150 mM NaCl, and 0.5% IGEPAL CA-630, and EDTA-free protease inhibitor cocktail (Roche, Basel, CH). The resin was then washed twice with a TN buffer containing 50 mM Tris (pH 7.5) and 150 mM NaCl. The resin was then frozen at –80°C for at least 1 hr. Proteins were eluted twice with shaking at 37°C for 1 hr with elution buffer (0.2 M glycine, 0.5% IGEPAL CA-630, pH 2.3). Elutions were immediately neutralized with 1 M ammonium bicarbonate.

MS sample preparation of co-IP samples was performed as described previously (*Kim et al., 2023*). Briefly, samples were precipitated in methanol/chloroform, washed three times with methanol, and air-dried. Protein pellets were then resuspended in 3 µl 1% Rapigest SF (Waters, Milford, MA). Samples were reduced with 5 mM TCEP (Sigma-Aldrich, St. Louis, MO), alkylated with 10 mM iodoacetamide (Sigma-Aldrich, St. Louis MO), and digested with 0.5 µg of Sequencing Grade trypsin (Promega, Madison, WI) overnight in 50 mM HEPES (pH 8.0) at 37°C with shaking. Digested peptides were labeled with TMT 18-plex reagents (Thermo Fisher, Pittsburg, PA). TMT-labeled samples were pooled and acidified with MS-grade formic acid (Sigma-Aldrich, St. Louis, MO) to remove cleaved Rapigest SF via centrifugation. Supernatant was concentrated using a SpeedVac and resuspended in buffer A (95% water, 4.9% acetonitrile, and 0.1% formic acid). The sample was then loaded onto a triphasic MudPIT column using a high-pressure chamber.

LC–MS/MS analysis was performed on an Exploris 480 mass spectrometer equipped with an Ulti-Mate3000 RSLCnano System (Thermo Fisher, Pittsburg, PA) as described previously (*McDonald et al., 2022a*). MudPIT experiments were performed with 10 µl sequential injections of 0, 10, 20, 40, 60, 80 and 100% buffer C (500 mM ammonium acetate in buffer A) were performed followed by a final injection of 90% buffer C with 10% buffer B (99.9% acetonitrile, 0.1% formic acid). Each step consisted of a 90-min gradient from 4 to 40% B with a flow rate of either 300 or 500 nl/min, followed by a 15-min gradient from 40 to 80% B with a flow rate of 500 nl/min on a 20-cm fused silica microcapillary column (ID 100 µm) ending with a laser-pulled tip filled with Aqua C18, 3 µm, 100 Å resin (Phenomenex, Torrance, CA). Electrospray ionization was performed directly from the analytical column by applying a voltage of 2.0 or 2.2 kV with an inlet capillary temperature of 275°C. Data-dependent acquisition of MS/MS spectra was performed by scanning from 300 to 1800 *m/z* with a resolution of 60,000–120,000. Peptides with an intensity above 1.0E4 with charge state 2–6 from each full scan were fragmented by higher energy collisional dissociation using normalized collision energy of 35–38 with a 0.4 *m/z* isolation window, 120 ms maximum injection time at a resolution of 45,000, scanned from 100 to 1800 *m/z* or defined a first mass at 110 *m/z*, and dynamic exclusion set to 45 or 60 s and a mass tolerance of 10 ppm.

Peptide identification and TMT-based protein quantification were carried out using Proteome Discoverer 2.4 (Thermo Fisher, Pittsburg, PA) as described previously (*Kim et al., 2023*). MS/MS spectra were extracted from Thermo XCaliber.raw file format and searched using SequestHT against a UniProt human proteome database (released 03/25/2014) containing 20,337 protein entries. The database was curated to remove redundant protein and splice isoforms and supplemented with common biological MS contaminants. Searches were carried out using a decoy database of reversed peptide sequences and the following parameters: 10 ppm peptide precursor tolerance, 0.02 Da fragment mass tolerance, minimum peptide length of six amino acids, trypsin cleavage with a maximum of two missed cleavages, static cysteine modification of 57.021 Da (carbamidomethylation), and static N-terminal and lysine modifications of 304.207 Da (TMT 18-plex). Search results were filtered using percolator to minimize the peptide false discovery rate to 1% and a minimum of two peptides per

protein identification. TMT reporter ion intensities were quantified using the reporter ion quantification processing node in Proteome Discoverer 2.4 and summed for peptides belonging to the same protein.

## Interactor filtering and data analysis

Six total TMT-18plex sets of samples were analyzed via LC–MS/MS over six separate mass spectrometry runs. The first three runs ($n = 4$ per condition) included WT (Parent), WT (CANX KO), ΔF508 (Parent), F508del (CANX KO), V232D (Parent), V232D (CANX KO), P5L (Parent), P5L (CANX KO), T1036N (Parent), T1036N (CANX KO), I1366N (Parent), and I1366N (CANX KO). A GFP control was included for both the parental and CANX KO cell lines ($n = 3$). The last three runs ($n = 6$ per condition) included ΔF508 (Parent) and F508del (CANX KO) treated with DMSO, 3 μM VX-445, 3 μM VX-661, or 3 μM VX-445 + 3 μM VX-661. Again, a GFP control was also included for both the parental and CANX KO cell lines ($n = 3$). Data from the first and last three runs were analyzed separately. Interactors co-immunoprecipitated with CFTR were filtered by comparing against a mock transfection control (GFP). The six separate $\log_2$ fold changes representing individual MS runs were then averaged to yield the consensus $\log_2$ fold change over mock values for each protein, and a paired two-tailed $t$-test was used to calculate the p-value.

To filter for true interactors of CFTR, a curved filter combining $\log_2$ fold change and p-value was used as described previously (**Davies et al., 2020**). Briefly, the histogram of $\log_2$ fold changes over mock was fit to a Gaussian curve using a nonlinear least-square fit to determine the SD (σ). Fold change cutoff for interactors was set to 2 σ. A reciprocal curve with the equation $y > c/(x − x_0)$, where $y$ = p-value, $x = \log_2$ fold change, $x_0$ = fold change cutoff (2 σ), and $c$ = the curvature ($c = 0.8$) was used to filter interactors in each condition. These interactors were pooled to generate a master list of true CFTR interactors. Raw abundances within runs were median normalized, followed by a bait normalization to CFTR. Briefly, each abundance value of an individual run was $\log_2$ transformed and averaged to yield the consensus $\log_2$-grouped abundance. The $\log_2$-grouped abundance values of proteins found in parental samples were then subtracted from the $\log_2$-grouped abundance values of proteins found in CANX KO samples to determine changes as a result of CANX loss.

Pathways were assigned by searching our dataset against the Proteostasis Consortium database (https://www.proteostasisconsortium.com). Novel interactors were assigned with pathways by searching the UniProt database. For violin plots, we used a one-sample $t$-test and a Wilcoxon signed-rank test to assess whether the log fold change of CANX KO relative to the parental cell line was significantly different from zero. To evaluate differences in aggregate pathway statistics across conditions, we performed a one-way ANOVA with Geisser–Greenhouse correction, followed by post hoc Tukey multiple comparisons testing.

## CFTR-mediated hYFP quenching measurements

CFTR function measurements were carried out in recombinant stable cells made from genetically modified HEK293T cells grown in 10 cm dishes in complete media as was previously described (**Carmody et al., 2024**). A pcDNA5 vector bearing an attB recombination site in place of its CMV promoter, untagged CFTR cDNA encoding each variant, and an IRES hYFP-P2A-mKate sensor cassette was used to generate a series of stable HEK293T cell lines that inducibly express each CFTR construct as previously described (**Carmody et al., 2024**). Recombinant HEK293T cells stably expressing CFTR were grown in six-well dishes and dosed with either DMSO (vehicle) or 3 μM VX-445 and VX-661 48 hr after plating. Following 16 hr. of treatment with correctors, the cells were harvested with trypsin, washed once in PBS containing 5 mM EDTA (pH 7.4), washed twice in PBS containing 137 mM sodium gluconate (pH 7.4), then resuspended in PBS containing 137 mM sodium gluconate (pH 7.4), 25 μM Forskolin, 50 μM Genistein, and 10 μM VX-770. Cells were incubated in the resuspension media for 10 min prior to a final wash and resuspension in PBS containing 137 mM sodium gluconate (pH 7.4) prior to analysis. hYFP quenching was initiated with the addition of 25 mM sodium iodide immediately prior to the analysis of single-cell forward light scattering, side light scattering, hYFP fluorescence (488 nm laser, 530/30 nm emission filter), and mKate fluorescence (561 nm laser, 610/20 nm emission filter) intensity values as a function of time using a BD Fortessa flow cytometer (BD Biosciences, Franklin Lakes, NJ). Data were analyzed using FlowJo software (Treestar, Ashland, OR). OriginPro

2019 (Origin, Northampton, MA) was used to globally fit the decay of single-cell hYFP: mKate ratios as a function of time using the following exponential decay function:

$$y = y_0 + A_1^{-k_t \cdot t},$$

where $y$ is the hYFP:mKate fluorescence intensity ratio at time $t$, $y_0$ is the baseline hYFP:mKate fluorescence intensity ratio at $t \to \infty$, $A_1$ is the amplitude of the signal, and $k_1$ is the observed rate constant for the exponential decay. The value of $y_0$ was fixed in the global fit to an experimentally determined value for the hYFP:mKate intensity ratio of cells prior to their treatment with sodium iodide. The quenching half-life was calculated from the globally fit parameters using the following equation:

$$t_{1/2} = \frac{\ln(2)}{k_1},$$

where $t_{1/2}$ is the half-life and $k_1$ is the globally fit rate constant.

## Short-circuit current (ISC) measurement

FRTs were grown on permeable filters for a total of 5 days to facilitate monolayer establishment and polarity. Forty-eight hours prior to functional and biochemical assessments, cells were treated basolaterally with CFTR correctors and/or potentiators (5 µM per drug). Cells were mounted in P2300 Ussing chambers (Physiologic Instruments, MC8 Apparatus) and analyzed for short-circuit currents as previously described (*Oliver et al., 2019*; *Rauscher et al., 2021*; *Sabusap et al., 2021*). Briefly, cells were equilibrated 5–10 min with basolateral regular Ringer buffer and apical low-chloride Ringer buffer. Amiloride (100 µM) was applied apically as a sodium channel inhibitor, after which the following CFTR agonists were added apically and/or basolaterally: forskolin (10 µM; activator of PKA) and VX-770 (5 µM; potentiator of CFTR gating). At the conclusion of each experiment, Inh172 (10 µM) was applied to the apical surface. CFTR-dependent transepithelial ion transport was measured as the difference (Δ) from baseline current to the highest or lowest value of a stable plateau achieved for 5–10 min.

## RNA extraction and quantitative reverse transcriptase-polymerase chain reaction

Steady-state mRNA levels were measured from FRT cells collected after short-circuit current analysis. Total RNA was isolated using a Direct-zol RNA Miniprep Plus Kit with DNase I (Zymo Research, Irvine, CA). RNA was then reverse transcribed using oligo-(dT)18 primers and Luna Probe One-Step RT-qPCR Mix (New England Biolabs, Ipswitch, MA) according to the manufacturer's instructions. Primers were purchased from IDT Technologies for human *CFTR* (Hs.PT.58.3365414), rat *CANX* (Rn. PT.58.34943747), and rat β-actin (*Actb*; Rn.PT.39a.22214838.g). These targets were amplified with the following primer sequences (5′ → 3′):

> *CFTR* forward, 5′-CCTTCGATATTTCACGCTCCA-3′, and reverse, 5′-GCCATTGTTTCCACCA TTAACG-3′;
> *CANX* forward, 5′-ACAAAGCTCCAGTTCCAACAG-3′, and reverse, 5′-TTTCATCTACCTCCCA CTTTCC-3′; and
> *Actb* forward, 5′-TCACTATCGGCAATGAGCG-3′, and reverse, 5′-GGCATAGAGGTCTTTACGGA TG-3′.

PCR was performed in sealed, clear, 96-well plates (Applied Biosystems, Waltham, MA) placed in a thermocycler (Applied Biosystems, 7500 Real Time PCR System). The ΔΔCT method was employed to calculate product levels of *CFTR* and *CANX* relative to *Actb*. Non-template and non-reverse transcribed samples served as negative controls.

## CFTR Rosetta comparative modeling and analysis

CFTR mutants were modeled using Rosetta comparative modeling as previously described (*McDonald et al., 2025*; *McDonald et al., 2022b*) Briefly, the five lowest-energy cryo-EM refined models (PDB ID

6MSM) were selected for VX-445 docking. Ligand conformations were generated with BCL Conformer-Generator, and VX-445 was aligned to published structures in Chimera before being docked 1000 times using RosettaLigand, with 24 cycles of full-atom docking and repacking. The final pose was chosen based on interface energy scores and root mean squared deviation (RMSD) from the published VX-445-bound structure (PDB ID 8EIQ). Mutations were introduced using Rosetta's MutateResidue mover.

Thermodynamic stability was assessed using Rosetta scores and Cα RMSD. Energy changes were calculated by comparing the Rosetta scores of apo and VX-445-bound ensembles. RMSD differences between mutant and wild-type (WT) CFTR were mapped onto CFTR structures. Mutation-induced instability was determined by subtracting the variant RMSD from WT RMSD, while VX-445 stabilization was assessed by comparing apo and VX-445-bound RMSD. These summed ΔRMSD values were quantified for NBD2 (residues 1201–1436). Finally, ΔRMSD was mapped onto the CFTR structure (PDB ID 6MSM) to investigate changes in local ensemble diversity interpreted as structural fluctuations.

## Acknowledgements

We thank Christiane Hassel and the Indiana University Flow Cytometry Core Facility for technical support. We thank Jill Hutchcroft at the Purdue University Flow Cytometry and Cell Separation Facility for technical support. We thank the Indiana University Center for Genomics and Bioinformatics for experimental support. This research was supported by grants from the National Institutes of Health (NIH) (R01HL167046 to JPS, LP, KEO, and JM). EFM was supported by a predoctoral fellowship from the National Heart, Lung, and Blood Institute (NHLBI) (F31 HL162483-01A1).

## Additional information

### Funding

| Funder | Grant reference number | Author |
| --- | --- | --- |
| National Heart, Lung, and Blood Institute | R01HL167046 | Jens Meiler<br>Kathryn E Oliver<br>Lars Plate<br>Jonathan P Schlebach |
| National Heart, Lung, and Blood Institute | F31HL162483-01A1 | Eli F McDonald |

The funders had no role in study design, data collection, and interpretation, or the decision to submit the work for publication.

### Author contributions

Austin Tedman, Conceptualization, Data curation, Formal analysis, Investigation, Visualization, Methodology, Writing – original draft, Writing – review and editing; John A Olson III, Data curation, Formal analysis, Investigation, Visualization, Methodology, Writing – review and editing; Minsoo Kim, Catherine Foye, JaNise J Jackson, Data curation, Formal analysis, Investigation, Writing – review and editing; Eli F McDonald, Data curation, Formal analysis, Investigation, Visualization, Writing – review and editing; Andrew G McKee, Formal analysis, Investigation, Methodology, Writing – review and editing; Karen Noguera, Investigation, Methodology, Writing – review and editing; Charles P Kuntz, Conceptualization, Data curation, Software, Formal analysis, Investigation, Methodology, Project administration, Writing – review and editing; Jens Meiler, Conceptualization, Resources, Software, Supervision, Methodology, Project administration, Writing – review and editing; Kathryn E Oliver, Conceptualization, Resources, Data curation, Formal analysis, Funding acquisition, Validation, Investigation, Visualization, Methodology, Project administration, Writing – review and editing; Lars Plate, Conceptualization, Resources, Data curation, Formal analysis, Supervision, Funding acquisition, Investigation, Methodology, Project administration, Writing – review and editing; Jonathan P Schlebach, Conceptualization, Resources, Data curation, Formal analysis, Supervision, Funding acquisition, Investigation, Visualization, Methodology, Writing – original draft, Project administration, Writing – review and editing

### Author ORCIDs
John A Olson III, https://orcid.org/0009-0000-9654-6611
Minsoo Kim https://orcid.org/0000-0001-9854-2999
Eli F McDonald https://orcid.org/0000-0002-0572-330X
Andrew G McKee https://orcid.org/0000-0003-1238-987X
Lars Plate https://orcid.org/0000-0003-4363-6116
Jonathan P Schlebach https://orcid.org/0000-0003-0955-7633

Reviewer #1 (Public review): https://doi.org/10.7554/eLife.107180.3.sa1
Reviewer #2 (Public review): https://doi.org/10.7554/eLife.107180.3.sa2
Author response https://doi.org/10.7554/eLife.107180.3.sa3

## Additional files

### Supplementary files
MDAR checklist

Supplementary file 1. CFTR variant read counts.

Supplementary file 2. CFTR expression table.

Supplementary file 3. CFTR primer list.

### Data availability
Code for the analysis of sequencing data for deep mutational scanning experiments can be found on the Schlebach Lab GitHub page (https://github.com/schebachlab/RP-dms-analysis copy archived at *schebachlab, 2022*). Illumina sequencing data is freely available through NCBI (https://www.ncbi.nlm.nih.gov/search/all/?term=PRJNA1244365). Fluorescence intensity values for mutants in all three mutational libraries are included as Excel files in the Supplementary Materials. The mass spectrometry proteomics data have been deposited to the ProteomeXchange Consortium via the PRIDE partner repository (PXD061664 and PXD066492). Structural modeling data and any associated code can also be accessed via GitHub (https://github.com/emcd173/CANX_KD_CFTR_Rosetta_Models copy archived at *McDonald, 2025*). All other raw experimental data have been deposited in a freely accessible Mendeley data directory (https://doi.org/10.17632/ksfmxv8zts.1).

The following datasets were generated:

| Author(s) | Year | Dataset title | Dataset URL | Database and Identifier |
|---|---|---|---|---|
| Schlebach J | 2025 | General Trends in the Calnexin-Dependent Expression and Pharmacological Rescue of Clinical CFTR Variants | https://www.ncbi.nlm.nih.gov/search/all/?term=PRJNA1244365 | NCBI, PRJNA1244365 |
| Plate L | 2025 | General Trends in the Calnexin-Dependent Expression and Pharmacological Rescue of Clinical CFTR Variants - DDA | https://www.ebi.ac.uk/pride/archive/projects/PXD061664 | PRIDE, PXD061664 |
| Tedman A, Olson J, Kim M, Foye C, Jackson J, McDonald E, McKee A, Noguera K, Kuntz C, Meiler J, Oliver K, Plate L, Schlebach J | 2025 | General Trends in the Calnexin-Dependent Expression and Pharmacological Rescue of Clinical CFTR Variants | https://doi.org/10.17632/ksfmxv8zts.1 | Mendeley Data, 10.17632/ksfmxv8zts.1 |
| Plate L | 2025 | General Trends in the Calnexin-Dependent Expression and Pharmacological Rescue of Clinical CFTR Variants – DIA | https://www.ebi.ac.uk/pride/archive/projects/PXD066492 | PRIDE, PXD066492 |

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
