## [Editor Report · eLife Assessment]

This **important** study advances our understanding of how cellular quality control machinery influences cystic fibrosis (CF) drug responsiveness by systematically analyzing the effects of the chaperone calnexin on more than two hundreds of CFTR (cystic fibrosis transmembrane regulator) variants. The evidence supporting the conclusions is **convincing**, with a comprehensive deep mutational scanning methodology and rigorous quantitative analysis. The findings reveal that calnexin is critical for both CFTR protein expression and corrector drug efficacy in a variant-specific manner, providing invaluable insights that could guide the development of personalized CF therapies. This work will be of significant interest to researchers in protein folding, CF drug development, and genetic disease therapeutics.

---

## [Referee Report · Reviewer #1 (Public review)]

Summary:

This research investigates how the cellular protein quality control machinery influences the effectiveness of cystic fibrosis (CF) treatments across different genetic variants. CF is caused by mutations in the CFTR gene, with over 1,700 known disease-causing variants that primarily work through protein misfolding mechanisms. While corrector drugs like those in Trikafta therapy can stabilize some misfolded CFTR proteins, the reasons why certain variants respond to treatment while others don't remain unclear. The authors hypothesized that the cellular proteostasis network-the machinery that manages protein folding and quality control-plays a crucial role in determining drug responsiveness across different CFTR variants. The researchers focused on calnexin (CANX), a key chaperone protein that recognizes misfolded glycosylated proteins. Using CRISPR-Cas9 gene editing combined with deep mutational scanning, they systematically analyzed how CANX affects the expression and corrector drug response of 234 clinically relevant CF variants in HEK293 cells.

In terms of findings, this study revealed that CANX is generally required for robust plasma membrane expression of CFTR proteins, and CANX disproportionately affects variants with mutations in the C-terminal domains of CFTR and modulates later stages of protein assembly. Without CANX, many variants that would normally respond to corrector drugs lose their therapeutic responsiveness. Furthermore, loss of CANX caused broad changes in how CF variants interact with other cellular proteins, though these effects were largely separate from changes in CFTR channel activity.

This study has some limitations: the research was conducted in HEK293 cells rather than lung epithelial cells, which may not fully reflect the physiological context of CF. Additionally, the study only examined known disease-causing variants and used methodological approaches that could potentially introduce bias in the data analysis.

How cellular quality control mechanisms influence the therapeutic landscape of genetic diseases is an emerging field. Overall, this work provides important cellular context for understanding CF mutation severity and suggests that the proteostasis network significantly shapes how different CFTR variants respond to corrector therapies. The findings could pave the way for more personalized CF treatments tailored to patients' specific genetic variants and cellular contexts.

Strengths:

(1) This work makes an important contribution to the field of variant effect prediction by advancing our understanding of how genetic variants impact protein function.

(2) The study provides valuable cellular context for CFTR mutation severity, which may pave the way for improved CFTR therapies that are customized to patient-specific cellular contexts.

(3) The research provides further insight into the biological mechanisms underlying approved CFTR therapies, enhancing our understanding of how these treatments work.

(4) The authors conducted a comprehensive and quantitative analysis, and they made their raw and processed data as well as analysis scripts publicly available, enabling closer examination and validation by the broader scientific community.

Comments on revisions:

The authors have addressed my concerns. If Document S1 is part of the final published version, this will address one of my previous concerns about potential skew and bias in the read data (Weakness 3, Methodological Choices).

---

## [Referee Report · Reviewer #2 (Public review)]

In this work, the authors use deep mutational scanning (DMS) to examine the effect of the endogenous chaperone calnexin (CANX) on the plasma membrane expression (PME) and potential pharmacological stabilization cystic fibrosis disease variants. This is important because there are over 1,700 loss-of-function mutations that can lead to the disease Cystic Fibrosis (CF), and some of these variants can be pharmacologically rescued by small-molecule "correctors," which stabilize the CFTR protein and prevent its degradation. This study expands on previous work to specifically identify which mutations affect sensitivity to CFTR modulators, and further develops the work by examining the effect of a known CFTR interactor-CANX-on PME and corrector response.

Overall, this approach provides a useful atlas of CF variants and their downstream effects, both at a basal level as well as in the context of a perturbed proteostasis. Knockout of CANX leads to an overall reduced plasma membrane expression of CFTR with CF variants located at the C-terminal domains of CFTR, which seem to be more affected than the others. This study then repeats their DMS approach, using PME as a readout, to probe the effect of either VX-445 or VX-455 + VX-661-which are two clinically relevant CFTR pharmacological modulators. I found this section particularly interesting for the community because the exact molecular features that confer drug resistance/sensitivity are not clear. When CANX is knocked out, cells that normally respond to VX-445 are no longer able to be rescued, and the DMS data show that these non-responders are CF variants that lie in the VX-445 binding site. Based on computational data, the authors speculate that NBD2 assembly is compromised, but that remains to be experimentally examined. Cells lacking CANX were also resistant to combinatorial treatment of VX-445 + VX-661, showing that these two correctors were unable to compensate for the lack of this critical chaperone.

One major strength of this manuscript is the mass spectrometry data, in which 4 CF variants were profiled in parental and CANX KO cells. This analysis provides some explanatory power to the observation that the delF508 variant is resistant to correctors in CANX KO cells, which is because correctors were found not to affect protein degradation interactions in this context. Findings such as this provide potential insights into intriguing new hypothesis, such as whether addition of an additional proteostasis regulators, such as a proteosome inhibitor, would facilitate a successful rescue. Taken together, the data provided can be generative to researchers in the field and may be useful in rationalizing some of the observed phenotypes conferred by the various CF variants, as well as the impact of CANX on those effects.

To complete their analysis of CF variants in CANX KO cells, the research also attempted to relate their data, primarily based on PME, to functional relevance. They observed that, although CANX KO results in a large reduction in PME (~30% reduction), changes in the actual activation of CFTR (and resultant quenching of their hYFP sensor) were "quite modest." This is an important experiment and caveat to the PME data presented above since changes in CFTR activity does not strictly require changes in PME. In addition, small molecule correctors also do not drastically alter CFTR function in the context of CANX KO. The authors reason that this difference is due to a sort of compensatory mechanism in which the functionally active CFTR molecules that are successfully assembled in an unbalanced proteostasis system (CANX KO) are more active than those that are assembled with the assistance of CANX. While I generally agree with this statement, it is not directly tested and would be challenging to actually test.

The selected model for all the above experiments was HEK293T cells. The authors then demonstrate some of their major findings in Fischer rat thyroid cell monolayers. Specifically, cells lacking CANX are less sensitive to rescue by CFTR modulators than the WT. This highlights the importance of CANX in supporting the maturation of CFTR and the dependence of chemical correctors on the chaperone. Although this is demonstrated specifically for CANX in this manuscript, I imagine a more general claim can be made that chemical correctors depend on a functional/balanced proteostasis system, which is supported by the manuscript data. I am surprised by the discordance between HEK293T PME levels compared to the CTFR activity. The authors offer a reasonable explanation about the increase in specific activity of the mature CFTR protein following CANX loss.

For the conclusions and claims relevant to CANX and CF variant surveying of PME/function, I find the manuscript to provide solid evidence to achieve this aim. The manuscript generates a rich portrait of the influence of CF mutations both in WT and CANX KO cells. While the focus of this study is a specific chaperone, CANX, this manuscript has the potential to impact many researchers in the broad field of proteostasis.

Comments on revisions:

The authors address my concerns. I appreciate seeing that the UPR probably isn't activated, ruling out that less PME is simply due to less CF protein.

---

## [Author Response]

The following is the authors’ response to the original reviews

**Reviewer 1 (Public review):**
This research investigates how the cellular protein quality control machinery influences the effectiveness of cystic fibrosis (CF) treatments across different genetic variants. CF is caused by mutations in the CFTR gene, with over 1,700 known disease-causing variants that primarily work through protein misfolding mechanisms. While corrector drugs like those in Trikafta therapy can stabilize some misfolded CFTR proteins, the reasons why certain variants respond to treatment while others don't remain unclear. The authors hypothesized that the cellular proteostasis network-the machinery that manages protein folding and quality control-plays a crucial role in determining drug responsiveness across different CFTR variants. The researchers focused on calnexin (CANX), a key chaperone protein that recognizes misfolded glycosylated proteins. Using CRISPR-Cas9 gene editing combined with deep mutational scanning, they systematically analyzed how CANX affects the expression and corrector drug response of 234 clinically relevant CF variants in HEK293 cells.In terms of findings, this study revealed that CANX is generally required for robust plasma membrane expression of CFTR proteins, and CANX disproportionately affects variants with mutations in the C-terminal domains of CFTR and modulates later stages of protein assembly. Without CANX, many variants that would normally respond to corrector drugs lose their therapeutic responsiveness. Furthermore, loss of CANX caused broad changes in how CF variants interact with other cellular proteins, though these effects were largely separate from changes in CFTR channel activity.This study has some limitations: the research was conducted in HEK293 cells rather than lung epithelial cells, which may not fully reflect the physiological context of CF. Additionally, the study only examined known diseasecausing variants and used methodological approaches that could potentially introduce bias in the data analysis.

We agree that the approaches employed here are not fully physiological, though we would remind the reviewer that we previously benchmarked the results generated by this experimental platform against a variety of other published datasets (PMID: 37253358). Regarding the issue of bias, we outline several pieces of evidence suggesting we retain robust and near-uniform sampling of these variants across these experimental conditions. We hope our comments below address all of these concerns. Overall, we believe deep mutational scanning is actually remarkably unbiased relative to other approaches due to the fact that all measurements are taken from a single dish of cells that is processed in parallel. Moreover, we show the trends are highly reproducible across replicates and users (see Figure S1).

How cellular quality control mechanisms influence the therapeutic landscape of genetic diseases is an emerging field. Overall, this work provides important cellular context for understanding CF mutation severity and suggests that the proteostasis network significantly shapes how different CFTR variants respond to corrector therapies. The findings could pave the way for more personalized CF treatments tailored to patients' specific genetic variants and cellular contexts.Strengths:(1) This work makes an important contribution to the field of variant effect prediction by advancing our understanding of how genetic variants impact protein function.(2) The study provides valuable cellular context for CFTR mutation severity, which may pave the way for improved CFTR therapies that are customized to patient-specific cellular contexts.(3) The research provides further insight into the biological mechanisms underlying approved CFTR therapies, enhancing our understanding of how these treatments work.(4) The authors conducted a comprehensive and quantitative analysis, and they made their raw and processed data as well as analysis scripts publicly available, enabling closer examination and validation by the broader scientific community.

We are grateful for this broad perspective on the general relevance of this work.

Weaknesses:(1) The study only considers known disease-causing variants, which limits the scope of findings and may miss important insights from variants of uncertain significance.

We agree with this caveat. A more comprehensive library of CFTR variants will undoubtedly be useful for assigning variants of uncertain significance, though we note that such a large library would involve trade-offs in depth/ coverage that will compromise the sensitivity/ precision of the measurements. This will, in turn, make it challenging to compare the effects of CFTR modulators across the spectrum of clinical variants. For this reason, we believe the current library will remain a useful tool for CF variant theratyping.

(2) The cellular context of HEK293 cells is quite removed from lung epithelia, the primary tissue affected in cystic fibrosis, potentially limiting the clinical relevance of the findings.

We concede this limitation, but note that we did carry out functional measurements in FRT monolayers, which are a prevailing model that closely mimics pharmacological outcomes in the clinic (see Fig. 6).

(3) Methodological choices, such as the expansion of sorted cell populations before genetic analysis, may introduce possible skew or bias in the data that could affect interpretation.

We respectfully disagree with this point. The recombination system we employ in these studies generates millions of recombinant cells per transfection, which corresponds to tens of thousands of clones per variant. Moreover, our sequencing data contain exhaustive coverage of every variant characterized herein within each of the final data sets. Generally, we do not see any evidence to suggest certain variants are lost from the population. We note that, while HEK293T cells are not the most physiological relevant system, they are robust to uniformly express these variants in a manner that provides a precise comparison of their effects and/ or response to CFTR modulators. To address this concern, we added Document S1 to the revised draft, which shows the total number of reads for each variant within each fraction and each experiment.

(4) While the impact on surface trafficking is convincingly demonstrated, how cellular proteostasis affects CFTR function requires further study, likely within a lung-specific cellular context to be more clinically relevant.

We agree with this caveat.

**Reviewer 1 (Recommendations for the authors):**
Major IssuesCell Growth Bias? After sorting cell populations into quartiles, cells were expanded before genetic analysis - if CFTR variants affect cell doubling time (e.g., severely misfolded variants causing cellular stress), this could skew variant abundance within sorted quartiles and bias results.

Based on several observations, we do not believe this to be a significant issue. First, we note that we previously benchmarked the quantitative outputs of these experiments against a variety of other investigations and found very good agreement with previous variant classifications and expression levels (PMID: 37253358). If there were significant bias, we believe this would have come up in our efforts to benchmark the assay. Second, we note that we typically create recombinant cell lines that express WT or ΔF508 CFTR only alongside each recombinant cellular library. Importantly, we have never observed any difference in the growth rate of cultures expressing different CFTR variants. Third, even if cells expressing certain variants grow slower, it seems likely this slow growth would consistently occur in the context of each sorted subpopulation. Given that scores are derived from the relative amount of identifications across each subpopulation, we do not suspect this should impact the scoring. Overall, we believe the robustness of this cell line is a key feature that allows us to avoid any such issues related to proteostatic toxicity.

(1) Please add methodological detail. The data analysis pipeline lacks adequate description beyond referencing prior studies - essential details about what the Plasma Membrane Expression (PME) values represent (fold enrichment vs input library) and calculation methods must be provided.

We thank the reviewer for this helpful comment. We have added the text below to the revised manuscript in order to provide more detail to the reader:

“Briefly, low quality reads that likely contain more than one error were first removed from the demultiplexed sequencing data. Unique molecular identifier sequences within the remaining reads were then counted within each sample to track the relative abundance of each variant. To compare read counts across fractions, the collection of reads within each population were then randomly down-sampled to ensure a consistent total read count across each sub-population. The surface immunostaining of each variant was then estimated by calculating the the weighted-average immunostaining intensity for each variant using the following equation:\begin{document}$$\displaystyle  < \mathrm I> _{{Variant}} = \frac{\sum_{k=0}^{4}< F > _k\, N_k} {\sum_{k=0}^{4} N_k}$$\end{document}

where ⟨I⟩_variant_ is the weighted-average fluorescence intensity of a given variant, ⟨F⟩_i_ is the mean fluorescence intensity associated with cells from the ith FACS quartile, and Ni is the number of variant reads in the i^th^ FACS quartile. Variant intensities from each replicate were normalized relative to one another using the mean surface immunostaining intensity of the entire recombinant cell population for each experiment to account for small variations in laser power and/ or detector voltage. Finally, to filter out any noisy scores arising from insufficient sampling, we repeated the down-sampling and scoring process then rejected any variant measurements that exhibit more than X% variation in their intensity scores across the two replicate analyses. The reported intensity values represent the average normalized intensity values from two independent down-sampling iterations across three biologicals replicates.”

(3) Add detail on library composition. The distribution of CFTR variants within the parental HEK293T library after landing pad insertion needs documentation, including any variant dropout or overrepresentation issues.

As noted in our previous work (PMID: 37253358), our CF variant library is quite uniform, with each mutant contributing on average, 0.43% of the library with a standard deviation of +/- 0.16%. This corresponds to an average read depth of over 40K reads per variant, per experimental condition in the final analyses. Indeed, the most abundant variant in the pool was ΔF508 (1.67% of total reads). In contrast, the least sampled variant was S549R (1647T>G) was still sampled an average of 3,688 times per replicate, which corresponds to 0.09% of the total reads. See Doc S1.

(4) Documentation of CFTR variant overlap between parental and CANX KO HEK293T libraries is needed, including whether every variant was present at equivalent input abundance in both libraries.

We thank the reviewer for this suggestion. Though there are small deviations in the composition of recombinant parental and knockout cell lines, the relative abundances of individual variants within the recombinant populations only differs by an average of 18.5% between the parental and knockout lines. There are no cases in which we observe a single variant increasing by more than 50% in the knockout line relative to the parent. However, there is a single variant, Y563N, that exhibits a 96% decrease in its abundance in the context of the knockout cell line. Nevertheless, even this variant was sampled over 1,000 times, and it’s final score passed all quality control metrics. In the revised draft, we have provided a complete table containing the total number of reads and percent of total reads for each variant for each cell line and condition (see Doc. S1).

(5) The section reporting CANX impact on functional rescue of CF variants requires clearer logic flow - the conclusion about higher specific activity of CFTR assembled without CANX appears misleading, given later discussion about CANX allowing suboptimally folded CFTR to traffic to the surface.

We apologize for any confusion. We invoked the term “specific activity” in the enzymological sense, which is to say the proportion of active enzyme (i.e. channel) at the plasma membrane differs in the knockout line. The logic is quite simple- if protein levels are lower while ion conductance remains the same in the knockout cells, then a higher proportion of the mature channels must be inactive in the parental cell line. Thus, we suspect fewer of the channels at the plasma membrane are active in the context of the parental cell line containing CANX. We considered modifications to the text in the discussion, but ultimately feel the current text strikes a reasonable balance between nuance and simplicity.

(6) In your discussion, consider that HEK293T cellular context differs significantly from lung epithelia, and the hYFP quenching assay may have insufficient dynamic range or high noise for detecting relevant functional differences.

We modified the following sentence in the discussion to introduce this possibility:

“While these discrepancies could stem from differences in the dynamic range of the functional assays, they may also suggest the stringency of QC is more finely tuned to ion channel biosynthesis in epithelial monolayers.”

Minor Issues(1) Include immunostaining quartiles as a supplementary figure overlaid on Figure 1A, and clarify whether quartiles were consistent across experiments or adjusted for each sort.

We added a new figure to demonstrate the gating approach in the revised manuscript (see Fig. S10). We have also added the following text to the Methods section:

“Sorting gates for surface immunostaining were independently set for each biological replicate and in each condition to ensure that the population was evenly divided into four equal subpopulations.”

(2) Figure 2C improvements. Flip the figure 180 degrees to position MSD1 and NBD1 on the left, replace the blue-to-red color scale with yellow-to-blue or monochromatic scaling for better intermediate value differentiation.

Respectfully, we prefer not to do this so that our figures can be easily compared across our previous and forthcoming publications. We chose this rendering because this view depicts certain trends in variant response more clearly.

(3) Indicate the location of ECL4 on the protein structure shown in Figure 2C for better reference.

We appreciate the suggestion. However, most of ECL4 is missing from the experimental cryo-EM models of CFTR due to a lack of density. For this reason, we did not modify the figure.

**Reviewer 2 (Public review):**
In this work, the authors use deep mutational scanning (DMS) to examine the effect of the endogenous chaperone calnexin (CANX) on the plasma membrane expression (PME) and potential pharmacological stabilization cystic fibrosis disease variants. This is important because there are over 1,700 loss-of-function mutations that can lead to the disease Cystic Fibrosis (CF), and some of these variants can be pharmacologically rescued by small-molecule "correctors," which stabilize the CFTR protein and prevent its degradation. This study expands on previous work to specifically identify which mutations affect sensitivity to CFTR modulators, and further develops the work by examining the effect of a known CFTR interactor-CANX-on PME and corrector response.Overall, this approach provides a useful atlas of CF variants and their downstream effects, both at a basal level as well as in the context of a perturbed proteostasis. Knockout of CANX leads to an overall reduced plasma membrane expression of CFTR with CF variants located at the C-terminal domains of CFTR, which seem to be more affected than the others. This study then repeats their DMS approach, using PME as a readout, to probe the effect of either VX-445 or VX-455 + VX-661-which are two clinically relevant CFTR pharmacological modulators. I found this section particularly interesting for the community because the exact molecular features that confer drug resistance/sensitivity are not clear. When CANX is knocked out, cells that normally respond to VX-445 are no longer able to be rescued, and the DMS data show that these non-responders are CF variants that lie in the VX-445 binding site. Based on computational data, the authors speculate that NBD2 assembly is compromised, but that remains to be experimentally examined. Cells lacking CANX were also resistant to combinatorial treatment of VX-445 + VX-661, showing that these two correctors were unable to compensate for the lack of this critical chaperone.One major strength of this manuscript is the mass spectrometry data, in which 4 CF variants were profiled in parental and CANX KO cells. This analysis provides some explanatory power to the observation that the delF508 variant is resistant to correctors in CANX KO cells, which is because correctors were found not to affect protein degradation interactions in this context. Findings such as this provide potential insights into intriguing new hypothesis, such as whether addition of an additional proteostasis regulators, such as a proteosome inhibitor, would facilitate a successful rescue. Taken together, the data provided can be generative to researchers in the field and may be useful in rationalizing some of the observed phenotypes conferred by the various CF variants, as well as the impact of CANX on those effects.To complete their analysis of CF variants in CANX KO cells, the research also attempted to relate their data, primarily based on PME, to functional relevance. They observed that, although CANX KO results in a large reduction in PME (~30% reduction), changes in the actual activation of CFTR (and resultant quenching of their hYFP sensor) were "quite modest." This is an important experiment and caveat to the PME data presented above since changes in CFTR activity does not strictly require changes in PME. In addition, small molecule correctors also do not drastically alter CFTR function in the context of CANX KO. The authors reason that this difference is due to a sort of compensatory mechanism in which the functionally active CFTR molecules that are successfully assembled in an unbalanced proteostasis system (CANX KO) are more active than those that are assembled with the assistance of CANX. While I generally agree with this statement, it is not directly tested and would be challenging to actually test.The selected model for all the above experiments was HEK293T cells. The authors then demonstrate some of their major findings in Fischer rat thyroid cell monolayers. Specifically, cells lacking CANX are less sensitive to rescue by CFTR modulators than the WT. This highlights the importance of CANX in supporting the maturation of CFTR and the dependence of chemical correctors on the chaperone. Although this is demonstrated specifically for CANX in this manuscript, I imagine a more general claim can be made that chemical correctors depend on a functional/balanced proteostasis system, which is supported by the manuscript data. I am surprised by the discordance between HEK293T PME levels compared to the CTFR activity. The authors offer a reasonable explanation about the increase in specific activity of the mature CFTR protein following CANX loss.For the conclusions and claims relevant to CANX and CF variant surveying of PME/function, I find the manuscript to provide solid evidence to achieve this aim. The manuscript generates a rich portrait of the influence of CF mutations both in WT and CANX KO cells. While the focus of this study is a specific chaperone, CANX, this manuscript has the potential to impact many researchers in the broad field of proteostasis.

We thank the reviewer for their thoughtful and comprehensive perspectives on the scope and relevance of this work.

**Reviewer 2 (Recommendations for the authors):**
While I did not identify any major weaknesses in this manuscript, I offer some suggestions below, as well as some conclusions to consider:(1) Missing period at the end of line 51.

We thank the reviewer for catching this grammatical error and have added proper punctuation.

(2)Figure S1 "repre-sent"??

We have corrected this punctuation error.

(3) Figure S2 missing parentheses A

We have corrected the punctuation error.

(4) Figure S5, "B The total ΔRMSD of the active conformation of NBD2 is shown for variants bound to VX-445. Red bars show increasing deviations from the native NBD2 conformation in the mutant models, and blue bars show how much VX-445 suppresses these conformational defects in NBD2."VX-445 should not bind/stabilize the G85E from the calculations in Figure S5A. As a confirmation, it would be nice to see the calculated hypothetical effect of VX-445 in the G85E variant as performed for L1077P and N1303K. I also want to point out that G58E is referred to as being non-responsive in S5A, but then in S5D, N103K is referred to as non-responsive, but this variant falls pretty far below the stabilized region calculated in S5A, right?

We agree that it would be insightful to examine the RMSD changes in a non-responsive variant such as G85E. We added the G85E NBD2 ∆RMSD to Supplemental Figure S5B and a G85E ∆RMSD structure map as an additional subpanel at Supplemental Figure S5C. As the reviewer expected, VX-445 fails to confer any stability to G85E as shown by a lack of significant change in NBD2 ∆RMSD or any visible ∆RMSD throughout the structure. Finally, we acknowledge that N1303K falls below the stabilized region as calculated in S5A. However, we note that the binding energy only suggests it is likely to interact with the protein- this does not to necessarily mean that binding will allosterically suppress conformational defects in NBD2. Moreover, this is simply an in silico calculation, that does not necessarily capture all of the nuanced interactions in the cell (or lack thereof). We have corrected this in the Figure S5 caption, which reads as follows:

“Maps of the change in RMSD between N1303K modeled with and without VX-445 shows that few structural regions are stabilized by VX-445 for N1303K, which responds poorly to VX-445 in vitro.”

(5) "stan-dard" standard?

We have corrected this punctuation error.

(6) Line 270, "these variants" is written twice

We have corrected this typographical error.

(7) Figure 6 B. What is being compared? The text writes "there are prominent differences in the activity of these variants [those with CANX] two-way ANOVA, p = 3.8 x 10-27." Does this mean WT vs. delF508, P5L, V232D, T1036N, and I1366N combined? I have not seen a set of 5 variables compared to a single variable. Usually, it would be WT vs. DelF508, WT vs. P5L, WT vs. V232D...right? Maybe this is normal in this specific field. The same goes for the CANX knockout comparison "(two-way ANOVA, p = 0.06).".

In this instance, the two-way ANOVA test is evaluating whether there are differences in the half-lives of individual variants and/ or systematic differences across the variant measurements in the knockout line relative to the parental cells. The test gives independent p-values for these two variables (variant and cell line). We chose this test because it makes it clear that, when you consider the trends together, one variable has a significant effect while the other does not.

(8) Why don't the CFTR modulators rescue CFTR activity in the WT FRT monolayers?

We thank the reviewer for this inquiry. Please note that compared to DMSO, VX-661 does significantly enhance the forskolin-mediated response of WT-CFTR (red asterisk). Treatments with VX-445 alone, VX-661+VX-445, or VX-661+VX-445+VX-770 showed no significant forskolin stimulation of WT-CFTR. These observations could be attributable to the brief period in which WT-CFTR cDNA is transiently transfected. However, it is not necessarily anticipated that modulators would enhance WT-CFTR function. Correctors and potentiators are designed to rescue processing and gating abnormalities, respectively. WT-CFTR channels do not exhibit such defects.

In both constitutive overexpression systems and primary human airway epithelia, published literature demonstrates that prolonged exposure to CFTR modulators has resulted in variable consequences on WT-CFTR activity. For example, forskolin-mediated responsiveness of WT-CFTR is not altered by chronic application of VX-445 (PMID: 34615919) nor VX-770 (PMID: 28575328, 27402691, 37014818). In contrast, short-circuit current measurements show that forskolin stimulation of WT-CFTR is augmented by chronic treatment with VX-809 (PMID: 28575328), an analog of VX-661. Thus, our findings are congruent with observations reported by other groups.

(9) General comment: As someone not familiar with the field, it would be nice to see the structures of VX-445 and VX-661 somewhere in the figures or at least in the SI.

We appreciate this suggestion, but do not feel that we include enough structural analyses to justify a stand-alone figure for these purposes. The structures of these compounds are easily referenced on a variety of internetbased resources.

(10) Weakness: As an ensemble, the data points CANX as required for plasma membrane expression, particularly those that lie in the C-terminal domain, but when considering individual CF variants, there is no clear trend. Similarly, when looking at the effect of the pharmacological correctors on PME, no variant strays from the linear trend.

We generally agree that the predominant trend is a uniform decrease in CFTR PME across all variants and that individual variant effects are hard to generalize. Indeed, this latter point has been widely appreciated in the CF community for several decades. Our approach exposes this variability in detail, but we concede that we cannot yet fully interpret the full complexity of the trends.

(11) Something to consider: Knockout of calnexin, a central ER chaperone, is going to set off the UPR, which in turn will activate the ISR and attenuate translation. From what I can tell, in general, all CF variant PME is decreased. Is this simply because less CF protein is being synthesized?

The reviewer raises an excellent point. However, to investigate this possibility further, we compared whole-cell proteomic data for the parental and knockout cell lines. Our analysis suggests there is no significant upregulation of proteins associated with UPR activation, as is shown in the graphic to the right. In fact, only proteins associated with the PERK branch of the UPR exhibit any statistically significant changes between these two cell lines across three biological replicates. Based on this consideration, we suspect any wider changes in ER proteostasis must be relatively subtle.
